# Maximization of Average Precision for Deep Learning with Adversarial Ranking Robustness

**Gang Li**
Texas A&M University
College Station, USA
`gang-li@tamu.edu`

**Wei Tong**
General Motors
Warren, USA
`wei.tong@gm.com`

**Tianbao Yang**
Texas A&M University
College Station, USA
`tianbao-yang@tamu.edu`

## Abstract

This paper seeks to address a gap in optimizing Average Precision (AP) while ensuring adversarial robustness, an area that has not been extensively explored to the best of our knowledge. AP maximization for deep learning has widespread applications, particularly when there is a significant imbalance between positive and negative examples. Although numerous studies have been conducted on adversarial training, they primarily focus on robustness concerning accuracy, ensuring that the average accuracy on adversarially perturbed examples is well maintained. However, this type of adversarial robustness is insufficient for many applications, as minor perturbations on a single example can significantly impact AP while not greatly influencing the accuracy of the prediction system. To tackle this issue, we introduce a novel formulation that combines an AP surrogate loss with a regularization term representing adversarial ranking robustness, which maintains the consistency between ranking of clean data and that of perturbed data. We then devise an efficient stochastic optimization algorithm to optimize the resulting objective. Our empirical studies, which compare our method to current leading adversarial training baselines and other robust AP maximization strategies, demonstrate the effectiveness of the proposed approach. Notably, our methods outperform a state-of-the-art method (TRADES) by more than 4% in terms of robust AP against PGD attacks while achieving 7% higher AP on clean data simultaneously on CIFAR10 and CIFAR100. The code is available at: `https://github.com/GangLii/Adversarial-AP`

## 1   Introduction

AP measures the precision of a model at different recall levels, offering a more nuanced understanding of the trade-offs between precision and recall. Optimizing AP for deep learning is of vital importance, especially in cases with highly imbalanced datasets. In such situations, accuracy alone can be misleading, as a model may perform well on the majority class but struggle with the minority class, thereby offering a superficially high accuracy score. In contrast, AP serves as a ranking metric that is particularly attuned to errors at the top of the ranking list, which makes it a more appropriate metric for applications dealing with highly imbalanced datasets. For example, deep AP maximization has been crucial in enhancing molecular property prediction performance, contributing to a winning solution in the MIT AICures challenge [54].

However, existing approaches of AP maximization are not robust against adversarial examples. It is notoriously known that deep neural networks (DNN) are vulnerable to adversarial attacks, where small, carefully-crafted perturbations to the input data can cause the model to produce incorrect predictions [46, 16]. These perturbations are often imperceptible to humans but can significantly impact the model's performance. Tremendous studies have been conducted to improve the adversarial robustness of DNN. A popular strategy to achieve adversarial robustness is through adversarial training [27, 59, 53, 45, 21, 2, 35, 44, 47, 15], which injects adversarial examples into the training

37th Conference on Neural Information Processing Systems (NeurIPS 2023).

Table 1: Comparison of different approaches for Adversarial AP Maximization (AdAP). Red indicates new features proposed in this paper.

| Approaches | Objective | Regularization | Optimization | Ranking Robustness | Trade-off | Consistent Attack |
|---|---|---|---|---|---|---|
| AdAP_MM | MiniMax AP Loss | No | Zero-sum Game | Yes | No | No |
| AdAP_PZ | AP Loss + Reg. | Pointwise | Zero-sum Game | No | Yes | Yes |
| AdAP_LZ | AP Loss + Reg. | Listwise | Zero-sum Game | Yes | Yes | No |
| AdAP_LN | AP Loss + Reg. | Listwise | Non-zero-sum Game | Yes | Yes | Yes |
| AdAP_LPN | AP Loss + Reg. | Listwise + Pointwise | Non-zero-sum Game | Yes | Yes | Yes |

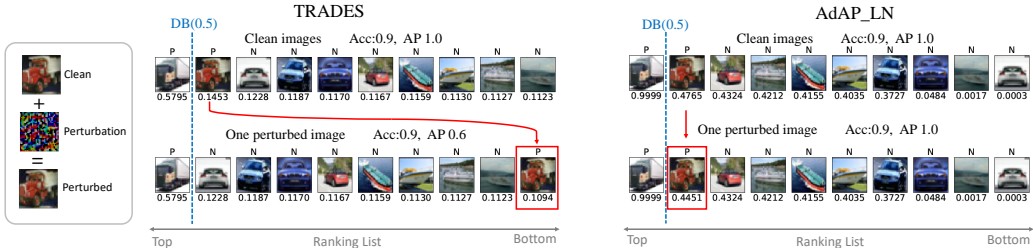

Figure 1: Top are predictions on clean images of a robust model trained by a state-of-the-art adversarial training method TRADES [59] and our method on CIFAR-10 data for detecting trucks. Bottom are predictions on the same set of images with only one example adversarially perturbed, which is generated by PGD following a black-box attack. The results of TRADES (left) indicate that slightly changing one example could dramatically impact AP but not on the accuracy. The results of our approach (right) demonstrate that our solution is more robust to adversarial data in terms of ranking and AP score. The dashed blue line indicates the decision boundary at score 0.5.

that are generated by various attack methods. Nevertheless, almost all existing methods focus on robustness concerning accuracy, ensuring that the average accuracy on adversarially perturbed examples is well maintained. This type of adversarial robustness is insufficient for many applications with highly imbalanced datasets, as minor perturbations on a single example can significantly impact AP while not greatly influencing the accuracy of the prediction system (cf. an example in Figure 1). This presents a significant challenge for adversarially robust AP maximization.

In this paper, we conduct a comprehensive study on how to imbue AP maximization with adversarial robustness. There are several technical and practical concerns in the design of adversarial training methods for AP maximization to enjoy three nice properties: (i) capability to trade off between AP on a set of clean data and the robustness of the model on the perturbed data; (ii) robustness in terms of ranking performance instead of accuracy against adversarial perturbations; (ii) consistency of attacks between training and inference. The first property is obvious in light of existing works on adversarial training [59]. The importance of the second property has been explained in the previous paragraph. The third property is tricky as it does not exist in existing adversarial training methods. The adversarial attack is usually applied to an individual data during the inference phase. Hence, we expect that maintaining the consistency between the attacks generated in training process and that in the inference phase will help boost the performance. However, this will cause a dilemma for achieving pointwise attack and listwise robustness in a unified framework.

To acquire these properties in a unified framework, we draw inspiration from prior adversarial training methods through robust regularization, and integrate two distinct design elements. We examine robust objectives that combine an AP surrogate loss on the clean data and a regularization term depending on the perturbed data. The two unique design features are (i) a new listwise adversarial regularization defined by a divergence between two distributions that represent the top one probabilities of a set of clean data and their perturbed versions; (ii) a non-zero-sum game approach, which integrates pointwise adversarial attacks with the proposed listwise adversarial regularization. This will ensure the attack consistency between training and inference. Our contributions are summarized below.

- We propose a non-zero-sum game optimization formulation for AP maximization with adversarial ranking robustness, which achieves listwise defense against pointwise attacks.

- We propose an efficient stochastic algorithm for solving the resulting objective, which integrates traditional adversarial sample generation methods with a state-of-the-art deep AP maximization algorithm without requiring a large batch size.

- We conduct extensive experiments to compare with existing leading adversarial training baselines, and ablation studies to compare different approaches shown in Table 1 for adversarial AP maxi-

mization. We conclude that AdAP_LN and AdAP_LPN achieve the best performance among all methods corroborating the effectiveness of the proposed techniques.

## 2 Related Work

**Adversarial Robustness.** To safeguard deep neural networks (DNNs) from adversarial attacks, a variety of adversarial defense techniques have been proposed in the literature, including (1) detection for defense [37, 48, 38, 24]; (2) input transformations for defense [25, 17, 39, 18, 58]; (3) adversarial training [1, 11, 47, 15]. Among these techniques, adversarial training has been demonstrated to be one of the most effective approaches [27, 59, 41, 45, 21, 14, 53]. Among these, [27] is the first to theoretically study and justify adversarial training by solving a min-max formulation for training adversarially robust models for deep learning. [59] presents an objective function that strike a balance between accuracy and robustness in light of their theoretical tight upper bound on the robust error. However, these previous methods focus on how to improve the robustness concerning accuracy, which is not sufficient for highly imbalanced data. [52] considers adversarial training for imbalanced data by combining a minimax weighted loss and a contrastive loss of feature representations. [60] considers adversarial ranking in the context of retrieval and proposes maximum-shift-distance attack that pushes an embedding vector as far from its original position as possible and uses it in a triplet loss for optimization. [20] presents a study on adversarial AUC optimization by reformulating the original tightly coupled objective as an instance-wise form for adversarial training. Nevertheless, none of these methods enjoy three nice properties simultaneously, i.e., adversarial ranking robustness, trade-off between AP and robustness, and consistent attacks between training and inference.

**Average Precision Optimization.** For imbalanced classification and information retrieval problems, AP optimization has attracted significant attention in the literature [13, 36, 31, 5, 10, 29, 28, 55]. To maximize the AP score for big data, some works employ stochastic optimization with mini-batch averaging to compute an approximate gradient of the AP function or its smooth approximation [6, 36, 42, 4]. These methods typically rely on a large mini-batch size for good performance. In contrast, [34] proposes a novel stochastic algorithm that directly optimizes a surrogate function of AP and provides theoretical convergence guarantee, without the need for a large mini-batch size. Then [50] further improve the stochastic optimization of AP by developing novel stochastic momentum methods with a better iteration complexity of $O(1/\epsilon^4)$. However, these approaches of AP maximization are vulnerable to adversarial examples created by introducing small perturbations to natural examples. The question of how to boost model's AP under adversarial perturbations while maintaining AP on clean data is still unresolved.

## 3 Preliminaries

For simplicity of exposition, we consider binary classification problems. However, the discussions and algorithms can be easily extended to mean AP for multi-class or multi-label classification problems. Let $\mathcal{D} = \{(\mathbf{x}_i, y_i)\}_{i=1}^n$ denote the set of all training examples with $\mathbf{x}_i$ being an input data and $y_i \in \{-1, 1\}$ being its associated label. Denoted by $h(\mathbf{x}) = h_\mathbf{w}(\mathbf{x})$ the predictive function (e.g., a deep neural network), whose parameters are $\mathbf{w} \in \mathbb{R}^d$. Denote by $\mathbb{I}(\cdot)$ an indicator function of a predicate. Denoted by $\|\cdot\| = \|\cdot\|_p$ the $L_p$-norm where $p \in (1, \infty]$. Denoted by $\mathbb{B}(0, \epsilon) = \{x : \|x\| \leq \epsilon\}$ the $L_p$-norm ball centered at 0 with radius $\epsilon$. Let $\mathcal{D}_+$ and $\mathcal{D}_-$ be the subsets of $\mathcal{D}$ with only positive examples and negative examples, respectively. Let $n_+ = |\mathcal{D}_+|$ denote the number of positive examples. Denote by $r(\mathbf{x}_i, \mathcal{D}) = \sum_{\mathbf{x}_j \in \mathcal{D}} \mathbb{I}(h_\mathbf{w}(\mathbf{x}_j) \geq h_\mathbf{w}(\mathbf{x}_i))$ the rank of $\mathbf{x}_i$ in a set $\mathcal{D}$ according to the prediction scores in descending order, i.e., the number of examples that are ranked higher than $\mathbf{x}_i$ including itself. Let $\ell(h_\mathbf{w}(\mathbf{x}), y)$ denote a pointwise loss function, e.g., the cross-entropy loss.

### 3.1 AP Maximization

According to its definition, AP can be expressed as: $\frac{1}{n_+} \sum_{\mathbf{x}_i \in \mathcal{D}_+} \frac{r(\mathbf{x}_i, \mathcal{D}_+)}{r(\mathbf{x}_i, \mathcal{D})}$ [3]. Many different approaches have been developed for AP maximization [9, 36, 13, 4, 34]. We follow a recent work [34], which proposes an efficient stochastic algorithm for AP maximization based on differentiable surrogate loss minimization. In particular, the rank function is approximated by using a surrogate loss of the indicator function $\mathbb{I}(h_\mathbf{w}(\mathbf{x}_j) \geq h_\mathbf{w}(\mathbf{x}_i))$ [34], yielding the following problem:

$$\min_\mathbf{w} P(\mathbf{w}) = -\frac{1}{n_+} \sum_{\mathbf{x}_i \in \mathcal{D}_+} \frac{\sum\limits_{s=1}^n \mathbb{I}(y_s = 1)\ell(\mathbf{w}; \mathbf{x}_s; \mathbf{x}_i)}{\sum\limits_{s=1}^n \ell(\mathbf{w}; \mathbf{x}_s; \mathbf{x}_i)}, \tag{1}$$

where $\ell(\cdot)$ denotes a smooth surrogate loss function, e.g., the squared hinge loss $\ell(\mathbf{w}; \mathbf{x}_s; \mathbf{x}_i) = (\max\{m - (h_\mathbf{w}(\mathbf{x}_i) - h_\mathbf{w}(\mathbf{x}_s)), 0\})^2$, where $m$ is a margin parameter.

To tackle the computational complexity of computing the stochastic gradient of $P(\mathbf{w})$, [34] has formulated this problem as a finite-sum coupled compositional optimization of the form $\frac{1}{n_+} \sum_{\mathbf{x}_i \in \mathcal{D}_+} f(\varphi(\mathbf{w}; \mathbf{x}_i))$, where $\varphi(\mathbf{w}; \mathbf{x}_i) = [\varphi_1(\mathbf{w}; \mathbf{x}_i), \varphi_2(\mathbf{w}; \mathbf{x}_i)]$, $\varphi_1(\mathbf{w}; \mathbf{x}_i) = \sum_{s=1}^{n} \mathbb{I}(y_s = 1)\ell(\mathbf{w}; \mathbf{x}_s; \mathbf{x}_i)/n$, $\varphi_2(\mathbf{w}; \mathbf{x}_i) = \sum_{s=1}^{n} \ell(\mathbf{w}; \mathbf{x}_s; \mathbf{x}_i)/n$, $f([\varphi_1, \varphi_2]) = -\varphi_1/\varphi_2$. To compute a stochastic gradient estimator, their algorithm maintains two moving average estimators $\mathbf{u}_{\mathbf{x}_i}^1$ and $\mathbf{u}_{\mathbf{x}_i}^2$ for $\varphi_1(\mathbf{w}; \mathbf{x}_i)$ and $\varphi_2(\mathbf{w}; \mathbf{x}_i)$. At iteration $t$, two mini-batches are sampeld $\mathcal{B}^+ \in \mathcal{D}_+$ and $\mathcal{B} \in \mathcal{D}$. The two estimators $\mathbf{u}_{\mathbf{x}_i}^1$ and $\mathbf{u}_{\mathbf{x}_i}^2$ are updated by Equation (2) for $\mathbf{x}_i \in \mathcal{B}^+$:

$$
\begin{aligned}
\mathbf{u}_{\mathbf{x}_i}^1 &= (1 - \gamma_1)\mathbf{u}_{\mathbf{x}_i}^1 + \gamma_1 \frac{1}{|\mathcal{B}|} \sum_{\mathbf{x}_j \in \mathcal{B}} \ell(\mathbf{w}_t; \mathbf{x}_j, \mathbf{x}_i)\mathbb{I}(y_j = 1) \\
\mathbf{u}_{\mathbf{x}_i}^2 &= (1 - \gamma_1)\mathbf{u}_{\mathbf{x}_i}^2 + \gamma_1 \frac{1}{|\mathcal{B}|} \sum_{\mathbf{x}_j \in \mathcal{B}} \ell(\mathbf{w}_t; \mathbf{x}_j, \mathbf{x}_i),
\end{aligned}
\tag{2}
$$

where $\gamma_1 \in (0, 1)$ is a moving average parameter. With these stochastic estimators, an stochastic estimate of $\nabla P(\mathbf{w})$ is given by Equation (3)

$$
\widehat{\nabla}_{\mathbf{w}_t} P(\mathbf{w}_t) = \frac{1}{|\mathcal{B}^+|} \sum_{\mathbf{x}_i \in \mathcal{B}^+} \sum_{\mathbf{x}_j \in \mathcal{B}} \frac{(\mathbf{u}_{\mathbf{x}_i}^1 - \mathbf{u}_{\mathbf{x}_i}^2 \mathbb{I}(\mathbf{y}_j = 1))\nabla\ell(\mathbf{w}_t; \mathbf{x}_j, \mathbf{x}_i)}{|\mathcal{B}|(\mathbf{u}_{\mathbf{x}_i}^2)^2}.
\tag{3}
$$

Then the model parameter can be updated similar to SGD, momentum-based methods, or Adam.

## 3.2 Pointwise Attacks and Pointwise Adversarial Regularization

An adversarial attack is usually applied to a specific example $\mathbf{x}$ such that its class label is changed from its original label (non-targeted attack) or to a specific label (targeted attack). We refer to this kind of attack as pointwise attack. Various pointwise attacking methods have been proposed, including, FGSM [16], Basic Iterative Method [23], PGD [27], JSMA [32], DeepFool [30], and CW attack [8]. While our approach is agnostic to any pointwise attacks, we restrict our discussion to optimization-based non-target attacks, $\delta = \arg\min_{\|\delta\| \leq \epsilon} G(\mathbf{w}, \mathbf{x} + \delta, y)$, where $G$ is an appropriate function, e.g., $G(\mathbf{w}, \mathbf{x} + \delta, y) = -\ell(h_\mathbf{w}(\mathbf{x} + \delta), y)$. A classic adversarial training method is to solve a robust optimization problem that integrates the training loss with the adversarial attack, e.g.,

$$
\min_{\mathbf{w}} \frac{1}{n} \sum_{i=1}^{n} \max_{\|\delta_i\| \leq \epsilon} \ell(h_\mathbf{w}(\mathbf{x}_i + \delta_i), y_i).
\tag{4}
$$

A deficiency of this approach is that it may not capture the trade-off between natural and robust errors [49]. To address this issue, robust regularization methods have been proposed, whose objective consists of a regular surrogate loss of error rate on clean data and a robust regularization term that accounts for the adversarial robustness. Different robust regularizations have been proposed [59, 45, 21]. A state-of-the-art approach is TRADES [59], whose objective is formulated by:

$$
\min_{\mathbf{w}} \frac{1}{n} \sum_{i=1}^{n} \left\{ \ell(h_\mathbf{w}(\mathbf{x}_i), y_i) + \lambda \max_{\|\delta_i\| \leq \epsilon} L(h_\mathbf{w}(\mathbf{x}_i), h_\mathbf{w}(\mathbf{x}_i + \delta_i)) \right\},
\tag{5}
$$

where $\lambda > 0$ is a regularization parameter and $L(\cdot, \cdot)$ is an appropriate divergence function, e.g., cross-entropy loss between two predicted probabilities [59].

## 4 Adversarial AP Maximization

First, we discuss two straightforward approaches. The first method (referred to as AdAP_MM in Table 1) is to replace the loss in (4) as an AP surrogate loss yielding the following:

$$
\min_{\mathbf{w}} \max_{\|\delta\| \leq \epsilon} -\frac{1}{n_+} \sum_{\mathbf{x}_i \in \mathcal{D}_+} \frac{\sum\limits_{s=1}^{n} \mathbb{I}(y_s = 1)\ell(\mathbf{w}, \mathbf{x}_s + \delta_s, \mathbf{x}_i + \delta_i)}{\sum\limits_{s=1}^{n} \ell(\mathbf{w}, \mathbf{x}_s + \delta_s, \mathbf{x}_i + \delta_i)}.
\tag{6}
$$

The second method (referred to as AdAP_PZ in Table 1) is to simply replace the first term in (5) with an AP surrogate loss:

$$
\min_{\mathbf{w}} P(\mathbf{w}) + \frac{\lambda}{n} \sum_{i=1}^{n} \max_{\|\delta_i\| \leq \epsilon} L(h_\mathbf{w}(\mathbf{x}_i), h_\mathbf{w}(\mathbf{x}_i + \delta_i)).
\tag{7}
$$

The limitations of the first method are that it may not capture the trade-off between AP and robustness and the adversarial attacks during the training are inter-dependent, which is not consistent with pointwise attacks generated during the inference phase. While the second method is able to trade off between AP and robustness, the pointwise regularization is not suitable for tackling imbalanced datasets, as an perturbation on a single data point from a minority class does not change the pointwise regularization too much but could degrade the AP significantly.

## 4.1 Listwise Adversarial Regularization

To address the deficiencies of the above straightforward approaches, we draw inspiration from TRADES and propose a listwise adversarial regularization to replace the pointwise regularization. The key property of the listwise adversarial regularization is that it should capture the divergence between the ranking result of the clean data and that of the perturbed data. To this end, we leverage the top-one probability proposed in the literature of learning to rank [7].

**Definition 1** *The top one probability of an data* $\mathbf{x}_i$ *represents the probability of it being ranked on the top, given the scores of all the examples i.e.,* $p_t(\mathbf{x}_i) = \frac{\exp(h_{\mathbf{w}}(\mathbf{x}_i))}{\sum_{j=1} \exp(h_{\mathbf{w}}(\mathbf{x}_j))}$.

With the definition of top-one probability, we define the listwise adversarial regularization as the divergence between two listwise probabilities, i.e., $\{p_t(\mathbf{x}_i)\}_{i=1}^n$ for the clean examples and $\{p_t(\mathbf{x}_i + \delta_i)\}_{i=1}^n$ for the perturbed examples. Different metrics can be used to measure the divergence between two distributions, e.g., KL divergence between $\{p_t(\mathbf{x}_i)\}_{i=1}^n$ and $\{p_t(\mathbf{x}_i + \delta_i)\}_{i=1}^n$, KL divergence between $\{p_t(\mathbf{x}_i + \delta_i)\}_{i=1}^n$ and $\{p_t(\mathbf{x}_i)\}_{i=1}^n$, and their symmetric version Jensen-Shannon divergence. For illustration purpose, we consider the KL divergence:

$$R(\mathbf{w}, \delta, \mathcal{D}) = \sum_{i=1}^n p_t(\mathbf{x}_i) \log p_t(\mathbf{x}_i) - p_t(\mathbf{x}_i) \log p_t(\mathbf{x}_i + \delta_i). \tag{8}$$

To further understand the above listwise regularization, we conduct a theoretical analysis similar to [59] by decomposing the robust error into two components. The difference from [59] is that we need to use the misranking error. We consider the misranking error as $\mathcal{R}_{nat} = \mathbb{E}_{\mathbf{x}_i \sim \mathcal{D}_+}\mathbb{I}\{h(\mathbf{x}_i) \leq \max_{\mathbf{x}_j \in \mathcal{D}_-} h(\mathbf{x}_j)\}$, which measures how likely a positive example is ranked below a negative example [43]. To characterize the robust ranking error under the attack of bounded $\epsilon$ perturbation, we define $\mathcal{R}_{rob} = \mathbb{E}_{\mathbf{x}_i \sim \mathcal{D}_+}\mathbb{I}\{\exists \delta_i, \delta_j \in \mathbb{B}(0, \epsilon), h(\mathbf{x}_i + \delta_i) \leq \max_{\mathbf{x}_j \in \mathcal{D}_-} h(\mathbf{x}_j + \delta_j)\}$. It is worth noting that $\mathcal{R}_{nat} \leq \mathcal{R}_{rob}$ is always satisfied, and in particular, $\mathcal{R}_{nat} = \mathcal{R}_{rob}$ if $\epsilon = 0$. We show that the $\mathcal{R}_{rob}$ can be decomposed into $\mathcal{R}_{nat}$ and $\mathcal{R}_{bdy}$ in Theorem 1.

**Theorem 1** $\mathcal{R}_{rob} = \mathcal{R}_{nat} + \mathcal{R}_{bdy}$, *where the second term is called the boundary error* $\mathcal{R}_{bdy} = \mathbb{E}_{\mathbf{x}_i \sim \mathcal{D}_+}\mathbb{I}\{h(\mathbf{x}_i) > \max_{\mathbf{x}_j \in \mathcal{D}_-} h(\mathbf{x}_j)\}\mathbb{I}\{\exists \delta_i, \delta_j \in \mathbb{B}(0, \epsilon), h(\mathbf{x}_i + \delta_i) \leq \max_{\mathbf{x}_j \in \mathcal{D}_-} h(\mathbf{x}_j + \delta_j)\}$.

**Remark:** It is clear that the boundary error measures the divergence between two ranked list $\{h_{\mathbf{w}}(\mathbf{x}_i)\}_{i=1}^n$ and $\{h_{\mathbf{w}}(\mathbf{x}_i + \delta_i)\}_{i=1}^n$, which provides an explanation of the divergence between the top-one probabilities on the clean data and on the perturbed data.

Since we are optimizing AP, we use an AP surrogate loss as a surrogate of the misranking error $\mathcal{R}_{nat}$ on the clean data. Finally, it yields in a robust objective $P(\mathbf{w}) + R(\mathbf{w}, \delta, \mathcal{D})$. A question remained is how to generate the perturbations. A simple strategy is to use the robust optimization approach that solves a zero-sum game: $\min_{\mathbf{w}} \max_{\delta \in \Omega^n} P(\mathbf{w}) + R(\mathbf{w}, \delta, \mathcal{D})$ (referred to as AdAP_LZ in Table 1). However, since $R(\mathbf{w}, \delta, \mathcal{D})$ is a listwise regularization, the resulting perturbations $\{\delta_i\}_{i=1}^n$ are inter-dependent, which is not consistent with the pointwise attacks generated in the inference phase. To address this issue, we decouple the defense and the attack by solving a non-zero-sum game:

$$\min_{\mathbf{w}} \quad P(\mathbf{w}) + \lambda R(\mathbf{w}, \delta, \mathcal{D})$$
$$\max_{\|\delta_i\| \leq \epsilon} \quad \sum_{i=1}^n G(\mathbf{w}, \mathbf{x}_i + \delta_i, y_i), \tag{9}$$

where the attacks are pointwise attacks generated for individual examples separately, e.g., FGSM, PGD. We refer to the above method as AdAP_LN. Finally, we experiment with another method by adding a pointwise regularization into the objective for optimizing the model parameter $\mathbf{w}$, i.e.,

$$\min_{\mathbf{w}} \quad P(\mathbf{w}) + \lambda(R(\mathbf{w}, \delta, \mathcal{D}) + \frac{1}{n}\sum_{i=1}^n L(h_{\mathbf{w}}(\mathbf{x}_i), h_{\mathbf{w}}(\mathbf{x}_i + \delta_i))) \tag{10}$$

This method referred to as AdAP_LPN imposes a stronger advesarial regularization on the model.

---

**Algorithm 1** Stochastic Algorithm for Solving AdAP_LN in (9)

---

1: Initialize $\mathbf{w}, \mathbf{u}_\mathbf{x}^1, \mathbf{u}_\mathbf{x}^2, \mathbf{u}, \gamma_1, \gamma_2$
2: **for** $t = 1, \ldots, T$ **do**
3:     Draw a batch of $B_+$ positive samples denoted by $\mathcal{B}_+$.
4:     Draw a batch of $B$ samples denoted by $\mathcal{B}$.
5:     **for** $\mathbf{x}_i \in \mathcal{B}$ **do**
6:         Initialize $\delta_i \sim \alpha \cdot \mathcal{N}(0, 1)$
7:         **for** $m = 1, \ldots, M$ **do**
8:             Update $\delta_i = \Pi_{\|\cdot\| \le \epsilon}(\delta_i + \eta_2 \cdot sign(\nabla_{\delta_i} G(\mathbf{w}_t, \mathbf{x}_i + \delta_i, y_i)))$, where $\Pi_\Omega(\cdot)$ is the projection operator.
9:         **end for**
10:     **end for**
11:     For each $\mathbf{x}_i \in \mathcal{B}_+$, update $\mathbf{u}_{\mathbf{x}_i}^1$ and $\mathbf{u}_{\mathbf{x}_i}^2$ by Equation (2)
12:     Update $\mathbf{u}$ by Equation (11) and compute $\widehat{\nabla}_{\mathbf{w}_t} R(\mathbf{w}_t, \delta, \mathcal{D})$ by Equation (12)
13:     Compute stochastic gradient estimator $\nabla_{\mathbf{w}_t} = \widehat{\nabla}_{\mathbf{w}_t} P(\mathbf{w}_t) + \lambda \widehat{\nabla}_{\mathbf{w}_t} R(\mathbf{w}_t, \delta, \mathcal{D})$
14:     Update $\mathbf{w}_{t+1}$ by using SGD, momentum-methods or Adam.
15: **end for**

---

## 4.2 Algorithms for Adversarial AP Maximization

Below, we will discuss efficient algorithms for adversarial AP maximization employing different objectives. Due to limit of space, we focus attention on solving (9). Equal efforts have been spent on solving other AdAP formulations with algorithms presented in the appendix.

Since there are two players one for minimizing over $\mathbf{w}$ and one for maximizing over $\delta$, we adopt the alternating optimization framework that first optimizes $\delta$ for generating attacks of sampled data and then optimizes for $\mathbf{w}$ based on the sampled clean data and perturbed data. The optimization for $\delta$ is following the existing methods in the literature. We use the PGD method for illustration purpose. The major technical challenge lies at computing a gradient estimator of the listwise regularization $R(\mathbf{w}, \delta, \mathcal{D})$ in terms of $\mathbf{w}$. We consider $\delta$ to be fixed and rewrite $R(\mathbf{w}, \delta, \mathcal{D})$ as:

$$\frac{\sum_{i=1}^n \exp(h(\mathbf{x}_i))(h(\mathbf{x}_i) - h(\mathbf{x}_i + \delta_i))}{\sum_{j=1}^n \exp(h(\mathbf{x}_j))} - \log \sum_{j=1}^n \exp(h(\mathbf{x}_j)) + \log \sum_{j=1}^n \exp(h(\mathbf{x}_j + \delta_j)).$$

Note that calculating the gradient of $R(\mathbf{w}, \delta, \mathcal{D})$ in terms of $\mathbf{w}$ directly, which includes the prediction scores of all samples, is not feasible. To tackle this challenge, we cast $R(\mathbf{w}, \delta, \mathcal{D})$ into a compositioal function and borrow the technique of stochastic compositional optimization. To this end, we define:

$$g(\mathbf{w}) = [g_1(\mathbf{w}), g_2(\mathbf{w}), g_3(\mathbf{w})]^\top$$
$$= \left[ \frac{1}{n} \sum_{i=1}^n \exp(h(\mathbf{x}_i))(h(\mathbf{x}_i) - h(\mathbf{x}_i + \delta_i)), \frac{1}{n} \sum_{i=1}^n \exp(h(\mathbf{x}_i)), \frac{1}{n} \sum_{i=1}^n \exp(h(\mathbf{x}_i + \delta_i)) \right]^\top.$$

Let $F(g) = \frac{g_1}{g_2} - \log g_2 + \log g_3$. Then we write $R(\mathbf{w}, \delta, \mathcal{D}) = F(g(\mathbf{w})) = \frac{g_1(\mathbf{w})}{g_2(\mathbf{w})} - \log g_2(\mathbf{w}) + \log g_3(\mathbf{w})$. The gradient of $R(\mathbf{w}; \delta, \mathcal{D})$ is given by:

$$\nabla_\mathbf{w} R(\mathbf{w}, \delta, \mathcal{D}) = \nabla_\mathbf{w} g(\mathbf{w})^\top \nabla F(g(\mathbf{w})) = \nabla_\mathbf{w} g(\mathbf{w})^\top \left( \frac{1}{g_2(\mathbf{w})}, \frac{-g_1(\mathbf{w}) - g_2(\mathbf{w})}{(g_2(\mathbf{w}))^2}, \frac{1}{g_3(\mathbf{w})} \right)^\top.$$

The major cost for computing $\nabla_\mathbf{w} R(\mathbf{w}, \delta, \mathcal{D})$ lies at evaluating $g(\mathbf{w})$ and its gradient $\nabla_\mathbf{w} g(\mathbf{w})$, which involves passing through all examples in $\mathcal{D}$. In order to develop a stochastic algorithm, we will approximate these quantities by using a mini-batch of random samples. The gradient $\nabla_\mathbf{w} g(\mathbf{w})$ can be simply approximated by the stochastic gradient, i.e.,

$$\widehat{\nabla}_\mathbf{w} g(\mathbf{w}) = \begin{pmatrix} \frac{1}{B} \sum_{\mathbf{x}_i \in \mathcal{B}} \exp(h(\mathbf{x}_i))((h(\mathbf{x}_i) - h(\mathbf{x}_i + \delta_i) + 1)\nabla_\mathbf{w} h(\mathbf{x}_i) - \nabla_\mathbf{w} h(\mathbf{x}_i + \delta_i)) \\ \frac{1}{B} \sum_{\mathbf{x}_i \in \mathcal{B}} \exp(h(\mathbf{x}_i))\nabla_\mathbf{w} h(\mathbf{x}_i) \\ \frac{1}{B} \sum_{\mathbf{x}_i \in \mathcal{B}} \exp(h(\mathbf{x}_i + \delta_i))\nabla_\mathbf{w} h(\mathbf{x}_i + \delta_i) \end{pmatrix},$$

where $\mathcal{B}$ denote a set of $B$ random samples from $\mathcal{D}$. Due to the non-linear dependence of $\nabla_\mathbf{w} R(\mathbf{w}, \delta, \mathcal{D})$ on $g(\mathbf{w})$, we cannot simply use their mini-batch estimator in the calculation as it will yield a large optimization error [51]. To reduce the optimization error [34, 33], we will maintain a vector $\mathbf{u} = [u_1, u_2, u_3]$ for tracking $[g_1(\mathbf{w}), g_2(\mathbf{w}), g_3(\mathbf{w})]$. The vector $\mathbf{u}$ is updated by Equation (11), where $\gamma_2 \in (0, 1)$ is a parameter. An estimate of $\nabla_\mathbf{w} R(\mathbf{w}, \delta, \mathcal{D})$ is given by

Equation (12):

$$\mathbf{u} = (1 - \gamma_2)\mathbf{u} + \gamma_2 \begin{pmatrix} \frac{\frac{1}{B}\sum_{\mathbf{x}_i \in \mathcal{B}} \exp(h(\mathbf{x}_i))(h(\mathbf{x}_i) - h(\mathbf{x}_i + \delta_i))}{\frac{1}{B}\sum_{\mathbf{x}_i \in \mathcal{B}} \exp(h(\mathbf{x}_i))} \\ \frac{1}{B}\sum_{\mathbf{x}_i \in \mathcal{B}} \exp(h(\mathbf{x}_i + \delta_i)) \end{pmatrix} \tag{11}$$

$$\widehat{\nabla}_{\mathbf{w}} R(\mathbf{w}, \delta, \mathcal{D}) = \widehat{\nabla}_{\mathbf{w}} g(\mathbf{w})^\top \left( \frac{1}{u_2}, \frac{-u_1 - u_2}{(u_2)^2}, \frac{1}{u_3} \right)^\top. \tag{12}$$

Finally, we are ready to present the detailed steps of the proposed algorithm for solving AdAP_LZ in (9) in Algorithm 1, which employs PGD for generating adversarial attacks and stochastic compositional optimization techniques for updating the model parameter $\mathbf{w}$.

## 5    Experiments

In this section, we perform extensive experiments to evaluate the proposed approaches against white-box and black-box attacks on diverse imbalanced datasets. In order to provide a comprehensive understanding of our methods, we conduct experiments from three other perspectives: (1) trade-off between robustness and average precision; (2) a close look at the effect of the adversarial perturbations; (3) ablation study of different strategies for adversarial AP maximization.

**Datasets.** We conduct experiments on four distinct datasets sourced from various domains. These encompass CIFAR-10 and CIFAR-100 datasets [22], CelebA dataset [26] and the BDD100K dataset [57]. For CIFAR-10 and CIFAR100, we adopt one versus-all approach to construct imbalanced classification tasks. It should be noted that even for clean data, the task of distinguishing between 1 positive class and 99 negative classes in CIFAR100 is challenging. Hence, in our experiments, we employ the 20 superclass labels, of which we choose the first 10 superclasses for constructing 10 one-vs-all binary classification tasks to verify the adversarial robustness on the CIFAR100 dataset. We split the training dataset into train/validation sets at 80%/20% ratio, and use the testing dataset for testing. CelebA dataset is a large-scale face attributes dataset that contains over 200,000 images of celebrities, each of which is annotated with 40 facial attributes. In our experiments, we choose two attributes with the highest imbalanced ratio, i,e., Gray_hair and Mustache, to show the superiority of our method. We adopt the recommended training/validation/testing split. BDD100K dataset is a large-scale diverse driving video dataset, which also has collected image-level annotation on six weather conditions and six scene types. In our experiments, we choose two kinds of weather conditions, i.e., Rainy and Cloudy, and two kinds of scene types, i.e., Tunnel and Residential, which are more imbalanced than others. Since the official testing dataset is not handy, we take the official validation set as the testing data, and split the training dataset into train/validation sets at 80%/20% ratio. The statistics for each task in datasets are presented in Table 6 in the appendix.

**Baselines.** In all experiments, we compare our methods (AdAP_LN, AdAP_LPN) with the following baseline methods: (1) PGD [27], which solves a MiniMax objective directly to enhance the adversarial robustness in terms of accuracy; (2) TRADES [59], which considers the trade-off between robustness and accuracy by minimizing a form of regularized surrogate loss; (3) MART [53], which explores the effect of misclassified examples to the robustness of adversarial training. Furthermore, we include two normal training methods, namely CE minimization and AP maximization by [34], as references.

**Experimental Details.** We employ the ResNet18 [19] as the backbone network in our experiments. This choice is based on the fact that all the tasks involved in our study are binary classifications, and ResNet18 is considered to be expressive enough for the purpose. For all methods, with mini-batch size as 128, we tune learning rate in {1e-3,1e-4,1e-5} with standard Adam optimizer. We set the weight decay to 2e-4 for the CIFAR10 and CIFAR100 datasets and 1e-5 for the CelebA and BDD100k datasets. In the case of the CIFAR10 and CIFAR100 datasets, we run each method for a total of 60 epochs. For the CelebA and BDD100k datasets, we run each method for 32 epochs. The learning rate decay happens at 50% and 75% epochs by a factor of 10. For MART, AdAP_LN and AdAP_LPN, we tune the regularization parameter $\lambda$ in {0.1, 0.4, 0.8, 1, 4, 8, 10}. For TRADES, we tune the regularization parameter in {1, 4, 8, 10, 40, 80, 100}, since they favor larger weights to obtain better robustness. In addition, for AdAP_LN and AdAP_LPN, we tune its moving average parameters $\gamma_1, \gamma_2$ in {0.1, 0.9}. Similarly, we tune the moving average parameters $\gamma_1$ for AP maximization in {0.1, 0.9}. We set margin parameter in the surrogate loss of AP as 0.6 for all methods that use the AP surrogate loss. For all adversarial training methods, we apply 6 projected gradient ascent steps to generate adversarial samples in the training stage, and the step size is 0.01. We choose $L_\infty$ norm to bound the perturbation within the limit of $\epsilon = 8/255$, as it is commonly used in the literature.

All the models in our experiments are trained from scratch, except direct AP maximization which is fine-tuned on a pretrained CE model as in [34]. For each experiment, we repeat three times with different random seeds, then report average and standard deviation.

## 5.1 Robustness against White-box Attacks

In this part, we evaluate the robustness of all models against PGD and APGD [12] white box attacks that have full access to model parameters. Specifically, we utilized 10-step PGD and APGD attack to generate adversarial perturbations constrained by the same perturbation limit $\epsilon = 8/255$. Given that APGD is a step size-free method, we set the step size for PGD to 0.007. For the adversarial training methods, hyperparameters and models are chosen based on the robust average precision (AP) metric on validation datasets, and their corresponding performances on test datasets are reported. For normal training methods, models are chosen based on the clean average precision. The results are presented in Table 2 and 3. Since we run all the classes of CIFAR10 and the first 10 classes of CIFAR100 to verify the effectiveness of our method, we report the mean average precision over the ten classes in the tables, and the performance on each class is shown in the appendix.

From Table 2 and 3, we can observe that our proposed methods outperform other baselines consistently by a large margin while maintaining notably higher clean AP scores. It is striking to see that our methods improve robust AP by $2 \sim 7$ percent on all datasets, compared with adversarial training methods which are concerning accuracy. The results also show that normal training methods (CE min. and AP max.) are vulnerable to adversarial attacks, while they can achieve pretty high clean AP. This is consistent with the observation from other literature [46, 16]. When comparing results in Table 3 with that in Table 2, we can see that APGD exhibits stronger attacks than PGD, as demonstrated in [12]. However, the superiority of our proposed methods still remains evident.

## 5.2 Robustness against Black-box Attacks

In this part, we examine all the models against adversarial black-box attacks. To achieve this, we utilize the model trained with CE loss minimization on clean datasets as an attacker model. So we do not include the performance of models of CE loss minimization here. With the well-trained model from CE loss minimization, we craft adversarial test images by PGD attack with 20 iterations and step size is 0.003, under perturbation limit $\epsilon = 8/255$. The models evaluated here are the same model evaluated in 5.1. The results are summarized in Table 4. Results show that our method exhibits significantly superior robustness against black-box attacks on all tasks, with evident advantages against white-box attacks shown in 5.1. From Table 4, we can also observe that the normal training method, i.e., AP maximization, is also susceptible when confronting adversarial black-box attacks.

## 5.3 Trade-off between Robustness and AP

As argued in various research studies [59, 40, 56], there exists a trade-off between accuracy and adversarial robustness. That is saying when one model boosts its adversarial performance, it may result in a decrease in its clean performance. In this part, we aim to study the trade-off between robustness and average precision and provide clear visualizations of the phenomenon. To accomplish this, we tune the weight of the regularization for the regularization-based approaches, i.e. TRADES, MART, AdAP_LN and AdAP_LPN, then show how their robust AP and clean AP changes. We evaluate the models at the last epoch to ensure that all methods reach a convergence point. For TRADES, AdAP_LN and AdAP_LPN, we tune the weight introduced in Experimental Details part, as well as 0.01, to better illustrate the trade-off trend. We also include non-regularization-based approaches as a point in this graph.

Based on Figure 2, we can observe that for TRADES, AdAP_LN and AdAP_LPN, as the weight of regularization increases, the clean AP decreases while the robust AP increases at first, which is consistent with the observation in [59]. But for MART, the trend is not clear as it is not a trade-off based approach. However, as the weight of regularization continues increasing, both the clean AP and robust AP decrease. This is because when the model places excessive emphasis on the regularization term, it may overlook the actual objective. Notably, our proposed methods place more towards the upper-right than other baselines, which indicates that our method is able to achieve better both robust and clean AP simultaneously.

## 5.4 Visualizing the Behavior after Adversarial Perturbations

To gain a deeper understanding of our approach, we have a close look at how defense models' predictions change after introducing adversarial perturbations. To be more specific, we compare our AdAP_LN method with another robust baseline, TRADES, to examine how their predictions are

Table 2: Adversarial robustness against PGD white box attacks

| Methods | CIFAR-10 (average) Robust | Clean | CIFAR-100 (average) Robust | Clean | CelebA(gray_hair) Robust | Clean | CelebA(mustache) Robust | Clean |
|---|---|---|---|---|---|---|---|---|
| CE Min. | 0.0526(0.0007) | 0.8944(0.0037) | 0.0256(0.0001) | 0.7094(0.0062) | 0.0161(0.0000) | 0.7851(0.0040) | 0.0196(0.0000) | 0.6658(0.0055) |
| AP Max. | 0.0528(0.0012) | 0.9007(0.0037) | 0.0267(0.0015) | 0.7222(0.0077) | 0.0161(0.0000) | 0.7851(0.0038) | 0.0196(0.0000) | 0.6517(0.0057) |
| PGD | 0.2765(0.0087) | 0.4929(0.0085) | 0.1613(0.0043) | 0.3149(0.0083) | 0.2576(0.0020) | 0.4946(0.0019) | 0.2127(0.0006) | 0.5040(0.0019) |
| TRADES | 0.2884(0.0029) | 0.5631(0.0045) | 0.1767(0.0026) | 0.3975(0.0065) | 0.2586(0.0024) | 0.4697(0.0040) | 0.2069(0.0016) | 0.5049(0.0001) |
| MART | 0.2859(0.0126) | 0.4971(0.0093) | 0.1650(0.0060) | 0.3167(0.0071) | 0.2608(0.0026) | 0.4874(0.0035) | 0.2147(0.0021) | 0.5053(0.0015) |
| AdAP_LN | 0.2949(0.0147) | 0.6345(0.0114) | 0.1957(0.0069) | 0.4628(0.0138) | **0.2931(0.0020)** | 0.5186(0.0030) | **0.2586(0.0044)** | 0.4693(0.0019) |
| AdAP_LPN | **0.3301(0.0022)** | 0.6526(0.0029) | **0.2188(0.0040)** | 0.4675(0.0075) | 0.2805(0.0028) | 0.4944(0.0047) | 0.2269(0.0010) | 0.5439(0.0040) |
| Methods | BDD100K(tunnel) Robust | Clean | BDD100K(residential) Robust | Clean | BDD100K(rainy) Robust | Clean | BDD100K(cloudy) Robust | Clean |
| CE Min. | 0.0014(0.0000) | 0.6289(0.0215) | 0.0655(0.0000) | 0.5970(0.0051) | 0.0379(0.0000) | 0.7374(0.0063) | 0.0379(0.0000) | 0.6623(0.0021) |
| AP Max. | 0.0014(0.0000) | 0.6390(0.0188) | 0.0656(0.0000) | 0.6262(0.0101) | 0.0381(0.0000) | 0.7566(0.0056) | 0.0388(0.0004) | 0.6681(0.0029) |
| PGD | 0.2263(0.0054) | 0.4091(0.0146) | 0.1378(0.0017) | 0.1805(0.0022) | 0.1730(0.0087) | 0.3688(0.0178) | 0.2164(0.0067) | 0.4497(0.0105) |
| TRADES | 0.1855(0.0094) | 0.4459(0.0208) | 0.1568(0.0011) | 0.2379(0.0012) | 0.2203(0.0016) | 0.4807(0.0023) | 0.2403(0.0013) | 0.5250(0.0028) |
| MART | 0.2223(0.0077) | 0.4074(0.0164) | 0.1400(0.0007) | 0.1865(0.0020) | 0.1703(0.0041) | 0.3661(0.0086) | 0.2157(0.0036) | 0.4469(0.0055) |
| AdAP_LN | 0.2918(0.0137) | 0.5726(0.0141) | 0.1617(0.0021) | 0.3070(0.0029) | 0.2545(0.0017) | 0.5401(0.0053) | **0.2826(0.0035)** | 0.5483(0.0039) |
| AdAP_LPN | **0.2986(0.0121)** | 0.5558(0.0047) | **0.1752(0.0013)** | 0.2552(0.0004) | **0.2598(0.0032)** | 0.5194(0.0033) | 0.2744(0.0036) | 0.5456(0.0087) |

Table 3: Adversarial robustness against APGD white-box attacks

| Methods | CIFAR-10 (average) Robust | Clean | CIFAR-100 (average) Robust | Clean | CelebA(gray_hair) Robust | Clean | CelebA(mustache) Robust | Clean |
|---|---|---|---|---|---|---|---|---|
| CE Min. | 0.0524(0.0004) | 0.8944(0.0037) | 0.0255(0.0000) | 0.7094(0.0062) | 0.0161(0.0000) | 0.7851(0.0040) | 0.0196(0.0000) | 0.6658(0.0055) |
| AP Max. | 0.0526(0.0012) | 0.9007(0.0037) | 0.0257(0.0003) | 0.7222(0.0077) | 0.0163(0.0001) | 0.7851(0.0038) | 0.0196(0.0000) | 0.6517(0.0057) |
| PGD | 0.2670(0.0054) | 0.4929(0.0085) | 0.1580(0.0035) | 0.3149(0.0083) | 0.2560(0.0020) | 0.4946(0.0019) | 0.2090(0.0004) | 0.5040(0.0019) |
| TRADES | 0.2839(0.0029) | 0.5631(0.0045) | 0.1739(0.0027) | 0.3975(0.0065) | 0.2572(0.0025) | 0.4697(0.0040) | 0.2020(0.0013) | 0.5049(0.0001) |
| MART | 0.2721(0.0052) | 0.4971(0.0093) | 0.1603(0.0050) | 0.3167(0.0071) | 0.2592(0.0025) | 0.4874(0.0035) | 0.2110(0.0021) | 0.5053(0.0015) |
| AdAP_LN | 0.2594(0.0056) | 0.6345(0.0114) | 0.1853(0.0030) | 0.4628(0.0138) | **0.2908(0.0009)** | 0.5186(0.0030) | **0.2318(0.0017)** | 0.4693(0.0019) |
| AdAP_LPN | **0.3252(0.0022)** | 0.6526(0.0029) | **0.2160(0.0040)** | 0.4675(0.0075) | 0.2785(0.0027) | 0.4944(0.0047) | 0.2222(0.0008) | 0.5439(0.0040) |
| Methods | BDD100K(tunnel) Robust | Clean | BDD100K(residential) Robust | Clean | BDD100K(rainy) Robust | Clean | BDD100K(cloudy) Robust | Clean |
| CE Min. | 0.0014(0.0000) | 0.6289(0.0215) | 0.0655(0.0000) | 0.5970(0.0051) | 0.0379(0.0000) | 0.7374(0.0063) | 0.0379(0.0000) | 0.6623(0.0021) |
| AP Max. | 0.0014(0.0000) | 0.6390(0.0188) | 0.0656(0.0000) | 0.6262(0.0101) | 0.0382(0.0001) | 0.7566(0.0056) | 0.0387(0.0003) | 0.6681(0.0029) |
| PGD | 0.2235(0.0055) | 0.4091(0.0146) | 0.1364(0.0013) | 0.1805(0.0022) | 0.1722(0.0086) | 0.3688(0.0178) | 0.2128(0.0067) | 0.4497(0.0105) |
| TRADES | 0.1819(0.0115) | 0.4459(0.0208) | 0.1563(0.0011) | 0.2379(0.0012) | 0.2187(0.0014) | 0.4807(0.0023) | 0.2366(0.0014) | 0.5250(0.0028) |
| MART | 0.2209(0.0083) | 0.4074(0.0164) | 0.1383(0.0013) | 0.1865(0.0020) | 0.1694(0.0040) | 0.3661(0.0086) | 0.2120(0.0038) | 0.4469(0.0055) |
| AdAP_LN | 0.2882(0.0142) | 0.5726(0.0141) | 0.1616(0.0013) | 0.3070(0.0029) | 0.2526(0.0019) | 0.5401(0.0053) | **0.2789(0.0027)** | 0.5483(0.0039) |
| AdAP_LPN | **0.2959(0.0114)** | 0.5558(0.0047) | **0.1747(0.0014)** | 0.2552(0.0004) | **0.2579(0.0031)** | 0.5194(0.0033) | 0.2719(0.0035) | 0.5456(0.0087) |

Table 4: Adversarial robustness against black-box attacks

| Methods | CIFAR-10 (average) Robust | Clean | CIFAR-100 (average) Robust | Clean | CelebA(gray_hair) Robust | Clean | CelebA(mustache) Robust | Clean |
|---|---|---|---|---|---|---|---|---|
| AP Max. | 0.1137(0.0374) | 0.9007(0.0037) | 0.0608(0.0241) | 0.7222(0.0077) | 0.0162(0.0000) | 0.7851(0.0038) | 0.0196(0.0000) | 0.6517(0.0057) |
| PGD | 0.4594(0.0074) | 0.4929(0.0085) | 0.2869(0.0078) | 0.3149(0.0083) | 0.4638(0.0022) | 0.4946(0.0019) | 0.4180(0.0010) | 0.5040(0.0019) |
| TRADES | 0.5148(0.0039) | 0.5631(0.0045) | 0.3420(0.0043) | 0.3975(0.0065) | 0.4360(0.0033) | 0.4697(0.0040) | 0.3953(0.0029) | 0.5049(0.0001) |
| MART | 0.4624(0.0083) | 0.4971(0.0093) | 0.2892(0.0067) | 0.3167(0.0071) | 0.4578(0.0039) | 0.4874(0.0035) | 0.4188(0.0013) | 0.5053(0.0015) |
| AdAP_LN | 0.5562(0.0087) | 0.6345(0.0114) | 0.3880(0.0099) | 0.4628(0.0138) | **0.4753(0.0011)** | 0.5186(0.0030) | 0.3635(0.0010) | 0.4627(0.0015) |
| AdAP_LPN | **0.6031(0.0028)** | 0.6526(0.0030) | **0.4166(0.0072)** | 0.4675(0.0075) | 0.4633(0.0047) | 0.4944(0.0047) | **0.4334(0.0036)** | 0.5439(0.0041) |
| Methods | BDD100K(tunnel) Robust | Clean | BDD100K(residential) Robust | Clean | BDD100K(rainy) Robust | Clean | BDD100K(cloudy) Robust | Clean |
| AP Max. | 0.1013(0.0736) | 0.6390(0.0188) | 0.1149(0.0367) | 0.6262(0.0101) | 0.0669(0.0228) | 0.7566(0.0056) | 0.1904(0.0694) | 0.6681(0.0029) |
| PGD | 0.3724(0.0123) | 0.4091(0.0146) | 0.1797(0.0022) | 0.1805(0.0022) | 0.3566(0.0172) | 0.3688(0.0178) | 0.4194(0.0089) | 0.4497(0.0105) |
| TRADES | 0.3764(0.0196) | 0.4459(0.0208) | 0.2330(0.0010) | 0.2379(0.0012) | 0.4645(0.0025) | 0.4807(0.0023) | 0.4752(0.0018) | 0.5250(0.0028) |
| MART | 0.3727(0.0161) | 0.4074(0.0164) | 0.1860(0.0020) | 0.1865(0.0020) | 0.3539(0.0085) | 0.3661(0.0086) | 0.4169(0.0047) | 0.4469(0.0055) |
| AdAP_LN | **0.5400(0.0168)** | 0.5726(0.0141) | **0.2872(0.0027)** | 0.3070(0.0029) | **0.5234(0.0052)** | 0.5401(0.0053) | 0.4872(0.0011) | 0.5483(0.0039) |
| AdAP_LPN | 0.5252(0.0062) | 0.5558(0.0047) | 0.2486(0.0008) | 0.2552(0.0004) | 0.5060(0.0037) | 0.5194(0.0033) | **0.4975(0.0085)** | 0.5457(0.0087) |

Table 5: Comparison of different adversarial AP maximization methods

| Methods | CIFAR-10 (average) Robust | Clean | CIFAR-100 (average) Robust | Clean | BDD100K(tunnel) Robust | Clean | BDD100K(rainy) Robust | Clean |
|---|---|---|---|---|---|---|---|---|
| AdAP_MM | 0.2759(0.0062) | 0.4708(0.0126) | 0.1878(0.0047) | 0.3843(0.0105) | 0.2743(0.0441) | 0.5849(0.0284) | 0.2199(0.0011) | 0.4516(0.0053) |
| AdAP_PZ | 0.3105(0.0018) | 0.6112(0.0049) | 0.2109(0.0032) | 0.4465(0.0078) | 0.2708(0.0180) | 0.5771(0.0516) | 0.2531(0.0029) | 0.5031(0.0104) |
| AdAP_LZ | 0.3012(0.0194) | 0.6508(0.0129) | 0.1919(0.0084) | 0.4422(0.0200) | 0.2619(0.0127) | 0.6212(0.0107) | 0.2103(0.0021) | 0.4365(0.0163) |
| AdAP_LN | 0.2949(0.0147) | 0.6345(0.0114) | 0.1957(0.0069) | 0.4628(0.0138) | 0.2918(0.0137) | 0.5726(0.0141) | 0.2545(0.0017) | 0.5401(0.0053) |
| AdAP_LPN | **0.3301(0.0022)** | 0.6526(0.0029) | **0.2188(0.0040)** | 0.4675(0.0075) | **0.2986(0.0121)** | 0.5558(0.0047) | **0.2598(0.0032)** | 0.5194(0.0033) |

affected by adversarial perturbations. For fair comparison, we apply the same perturbation crafted in 5.2 for both methods. From Figure 3, we can observe that for TRADES, all the samples' prediction scores cluster closely to a small value before perturbation as it is working on an imbalanced dataset, while the prediction scores of our method scatter more evenly. With introduced perturbations, the ranking order predicted by TRADE is changed significantly with small changes in prediction scores, which leads to a considerable decrease in terms of AP. On the contrary, the perturbation does not affect the predicted ranking order of our method, despite the impact on their prediction scores. This helps demonstrate why our method is more robust than other baselines in terms of AP.

## 5.5 Ablation Study

To investigate the significance of the new features in our proposed method, i.e., the listwise adversarial regularization and the non-zero-sum game approach, we conduct ablation studies to compare different approaches shown in Table 1 for adversarial AP maximization. Following the setting in 5.1, we

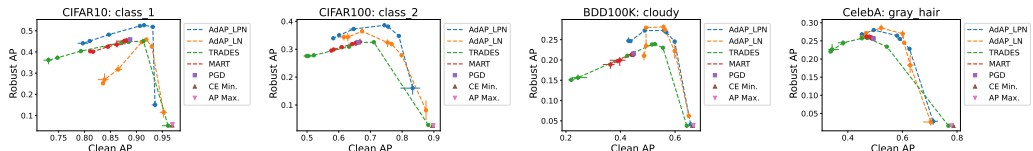

Figure 2: Visualization of the trade-off between robustness and average precision. For the sake of clarity in these figures, we defer the detailed values to Table 9 in the appendix to additionally demonstrate the correlation between robust AP/clean AP and $\lambda$.

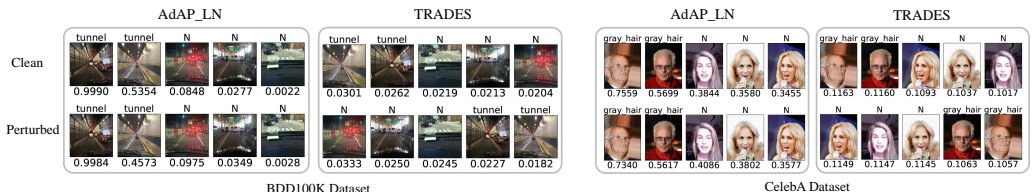

Figure 3: Visualization of models' predictions after perturbation

run all these five approaches on Cifar10, CIFAR100 and BDD100K datasets. The experimental settings and the hyperparameters for AdAP_LN and AdAP_LPN are the same as those in 5.1. The hyperparameters we tune for AdAP_LZ are the same as those for AdAP_LN. For AdAP_PZ, we tune the weight parameter in {1, 4, 8, 10, 40, 80, 100}. For AdAP_MM, we tune $\gamma_1$ in {0.1,0.9}. The results are presented in Table 5. We can see that (i) robust regularization approaches are better than the minimax approach (AdAP_MM) (ii) the non-zero-sum game approaches (AdAP_LPN, AdAP_LN) are usually better than the zero-sum game approach (AdAP_LZ); (iii) combining listwise and pointwise adversarial regularization in AdAP_LPN could bring significant boost in term of both robust AP and clean AP.

**Insensitivity to Batch Size.** We investigate the proposed approximation method for $g(\mathbf{w})$ by varying the mini-batch size for AdAP_LN algorithm and report results in Figure 4 in the appendix. We can see that AdAP_LN does not require a very large batch size and is generally not sensitive to the mini-batch size, which implies the effectiveness of our approximation method.

**Sensitivity to $\gamma_1, \gamma_2$.** We study the sensitivity of the hyper-parameters $\gamma_1, \gamma_2$ for proposed AdAP_LN algorithm. From the results in Table 8 in the appendix, we can observe that $\gamma_1, \gamma_2$ have a significant impact on the performance. However, when we tune $\gamma_1, \gamma_2$ in {0.1, 0.9}, it is able to achieve relatively good performance but not always the optimal one.

**Empirical convergence analysis.** We report the convergence curves of proposed AdAP_LN algorithm in Figure5 in the appendix, which demonstrates the convergence of the algorithm.

**Time efficiency.** We compare the training time efficiency of the proposed method with different algorithms in Table 7 in the appendix. We can observe that (i) adversarial training methods are generally more time-consuming than natural training; (ii) our proposed AdAP_LN and AdAP_LPN methods cost a little more time than traditional PGD method but much less time than TRADES.

## 6 Conclusions

In this paper, we have proposed a novel solution for maximizing average precision with adversarial ranking robustness. The proposed formulation is robust in terms of listwise ranking performance against individual adversarial perturbations and is able to trade off between average precision on clean data and adversarial ranking robustness. We have developed an efficient stochastic algorithm to solve the resulting objective. Extensive experimental results on various datasets demonstrate that the proposed method can achieve significantly improved adversarial robustness in terms of AP, compared with other strong adversarial training baselines. It would be interesting to extend our method to construct more robust systems with average precision as the objective, such as object detection systems and medical diagnosis systems.

## Acknowledgements

We thank anonymous reviewers for constructive comments. G. Li and T. Yang were partially supported by NSF Career Award 2246753, NSF Grant 2246757, NSF Grant 2246756, NSF Grant 2306572, and GM gift funding.

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

# A    More Experimental Results

All experiments in our paper are run across 16 NVIDIA A10 GPUs and 10 NVIDIA A30 GPUs. We present the statistics for each task in our experiments in Table 6.

Table 6: Datasets statistics. The percentage in parenthesis represents the proportion of positive samples.

| Dataset | Train | Validation | Test |
|---|---|---|---|
| CIFAR-10 | 40000 (10%) | 10000 (10%) | 10000 (10%) |
| CIFAR-100 | 40000 (5%) | 10000 (5%) | 10000 (5%) |
| BDD100K(tunnel) | 55890 (0.18%) | 13973 (0.19%) | 10000 (0.27%) |
| BDD100K(residential) | 55890 (11.56%) | 13973 (11.56%) | 10000 (12.53%) |
| BDD100K(rainy) | 55890 (7.26%) | 13973 (7.26%) | 10000 (7.38%) |
| BDD100K(cloudy) | 55890 (6.99%) | 13973 (6.98%) | 10000 (7.38%) |
| CelebA(gray_hair) | 162770 (4.24%) | 19867 (4.87%) | 19962 (3.19%) |
| CelebA(mustache) | 162770 (4.08%) | 19867 (5.05%) | 19962 (3.87%) |

## A.1    Time efficiency of our proposed algorithm

Time efficiency is typically a concern for adversarial training techniques. To order to investigate the time efficiency of our proposed method, we've conducted some experiments to show time efficiency comparison with different algorithms. In the experiment, we set the parameters, which could affect training time, exactly the same(e.g. batch size as 128, total epochs as 60, adversarial samples are generated with 6 projected gradient ascent steps) and run all the models over three times on the Class_0 task of CIFAR10. From the table 7, we can observe that (i) adversarial training methods are generally more time-consuming than natural training; (ii) our proposed AdAP_LN and AdAP_LPN methods cost a little more time than traditional PGD method but much less time than TRADES. This is because, to generate adversarial samples in training, TRADES is solving the maximization of KL divergence between the probabilities predicted with clean data and perturbed data(i.e. $\max_{\|\delta\| \leq \epsilon} \sum_k h(x)_k \log h(x)_k - h(x)_k \log h(x+\delta)_k$, where $h(x)_k$ and $h(x+\delta)_k$ are predicted probabilities for class k on clean data and perturbed data respectively, which requires two forward propagations and one backpropagation in each projected gradient ascent step. However, PGD and our proposed methods are directly solving maximization of Cross Entropy(i.e. $\max_{\|\delta\| \leq \epsilon} -\log h(x+\delta)_y$), which only needs one forward propagation and one backpropagation in each projected gradient ascent step. In adversarial training, since each gradient descent step w.r.t $\mathbf{w}$ requires multiple gradient ascent steps w.r.t $\delta$, the computational expense primarily stems from the projected gradient ascent steps, which can be also observed by comparing the efficiency of CE Min. with PGD.

Table 7: Training efficiency comparison

| Methods | Run 1 | Run 2 | Run 3 | Average |
|---|---|---|---|---|
| CE Min. | 563s | 566s | 568s | 565.67s |
| AP Max. | 589s | 590s | 589s | 589.33s |
| PGD | 2833s | 2804s | 2803s | 2813.33s |
| TRADES | 4203s | 4182s | 4179s | 4188.00s |
| MART | 3192s | 3205s | 3194s | 3197.00s |
| AdAP_LN | 3227s | 3213s | 3211s | 3217.00s |
| AdAP_LPN | 3234s | 3219s | 3218s | 3223.67s |

## A.2    Sensitivity of proposed algorithm to batch size

In our proposed algorithm, one essential step is the approximation of $g(\mathbf{w})$. Due to the non-linear dependence of $\nabla_{\mathbf{w}} R(\mathbf{w}, \delta, \mathcal{D})$ on $g(\mathbf{w})$, we cannot simply use their mini-batch estimator in the calculation as it will yield a large optimization error [51]. Instead, we maintain a vector $\mathbf{u} = [u_1, u_2, u_3]$ for tracking $[g_1(\mathbf{w}), g_2(\mathbf{w}), g_3(\mathbf{w})]$ to reduce the optimization error. The vector $\mathbf{u}$ is updated by Equation (11). To offer more empirical results on this, we have conducted some experiments to investigate the proposed AdAP_LN's sensitivity to batch sizes on CIFAR10 dataset. The results are shown in Figure 4. We can observe that our method does not require a very large batch size to achieve a good performance and is generally not sensitive to batch size, which implies the effectiveness of our approximation method.

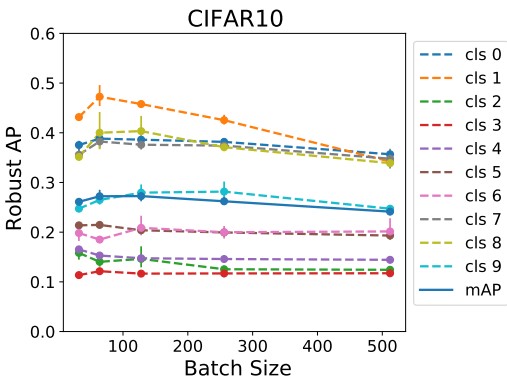

Figure 4: Illustration for insensitivity of AdAP_LN to batch size

## A.3 Sensitivity of proposed algorithm to $\gamma_1$ and $\gamma_2$

In our main experiments, the hyper-parameters $\gamma_1, \gamma_2$ for our proposed algorithms are tuned in $\{0.1, 0.9\}$ as mentioned in Section 5. To study the sensitivity of these parameters, we conducted some more experiments on CIFAR10 and BDD100K datasets. The results are summarized in Table 8. From the table, we can observe that $\gamma_1, \gamma_2$ have a significant impact on the performance. However, when we tune $\gamma_1, \gamma_2$ in $\{0.1, 0.9\}$, it is able to achieve relatively good performance but not always the optimal one.

Table 8: Robust AP on validation for AdAP_LN with different $\gamma_1$ and $\gamma_2$

| CIFAR10 (average) | $\gamma_2 = 0.1$ | $\gamma_2 = 0.3$ | $\gamma_2 = 0.5$ | $\gamma_2 = 0.7$ | $\gamma_2 = 0.9$ |
|---|---|---|---|---|---|
| $\gamma_1 = 0.1$ | 0.2719(0.0116) | 0.273(0.0079) | 0.275(0.0105) | 0.2756(0.0135) | 0.2753(0.0114) |
| $\gamma_1 = 0.3$ | 0.2701(0.012) | 0.2696(0.01) | 0.2753(0.0128) | 0.2733(0.0101) | 0.2745(0.0107) |
| $\gamma_1 = 0.5$ | 0.2674(0.0088) | 0.2741(0.011) | 0.2688(0.0108) | 0.2685(0.0135) | 0.2741(0.0113) |
| $\gamma_1 = 0.7$ | 0.2649(0.0103) | 0.2713(0.007) | 0.2686(0.0165) | 0.2696(0.0138) | 0.2658(0.0103) |
| $\gamma_1 = 0.9$ | 0.2668(0.0137) | **0.2766(0.0131)** | 0.2668(0.013) | 0.2689(0.0139) | 0.2641(0.0124) |
| BDD100K(rainy) | $\gamma_2 = 0.1$ | $\gamma_2 = 0.3$ | $\gamma_2 = 0.5$ | $\gamma_2 = 0.7$ | $\gamma_2 = 0.9$ |
| $\gamma_1 = 0.1$ | 0.2433(0.0209) | 0.2436(0.0205) | 0.2404(0.0188) | 0.2415(0.0220) | 0.2423(0.0215) |
| $\gamma_1 = 0.3$ | 0.2471(0.0219) | 0.2440(0.0174) | 0.2432(0.0169) | 0.2473(0.0214) | 0.2460(0.0188) |
| $\gamma_1 = 0.5$ | 0.2479(0.0210) | 0.2475(0.0209) | 0.2476(0.0217) | 0.2450(0.0189) | 0.2472(0.0187) |
| $\gamma_1 = 0.7$ | 0.2507(0.0203) | 0.2453(0.0204) | 0.2471(0.0179) | 0.2461(0.0206) | 0.2480(0.0176) |
| $\gamma_1 = 0.9$ | **0.2522(0.0208)** | 0.2479(0.0200) | 0.2485(0.0201) | 0.2447(0.0193) | 0.2478(0.0193) |

## A.4 Empirical convergence analysis

Since the theoretical convergence analysis of our proposed algorithm is very challenging, deriving the convergence would be a significant work by itself. Instead, we provides some empirical results to show the convergence of the proposed algorithm. Specifically, we set $\lambda = 1, \gamma_1 = 0.1, \gamma_2 = 0.1$ and run AdAP_LN algorithm on CIFAR10 dataset and BDD100K dataset for a total of 120 epochs and 80 epochs, respectively. We evaluate the training loss after each epoch and report the loss values. We present the AP loss (i.e., $P(\mathbf{w})$ in Equation 9) and regularization term (i.e., $R(\mathbf{w}, \delta, \mathcal{D})$ in Equation 9) separately, as well as the summation of the two losses. For each experiment, we repeat three times with different random seeds. The results are shown in Figure 5, which demonstrates the convergence of our algorithm.

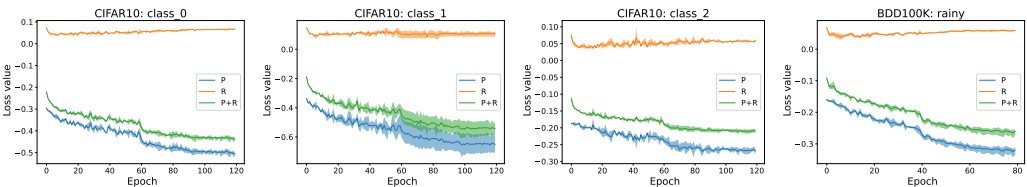

Figure 5: Training convergence curves on CIFAR10 and BDD100K dataset

## A.5 Detailed values for Figure 2

To clearly demonstrate the relationship between robust AP/clean AP and $\lambda$, corresponding to Fig.2, we provide the detailed values in Table 9. For TRADES, AdAP_LN and AdAP_LPN, the results clearly indicate that there is a positive correlation between robust AP and $\lambda$ for relatively small values of $\lambda$ and this correlation diminishes when $\lambda$ continues increasing. Moreover, a negative correlation between clean AP and $\lambda$ is consistently observed, in accordance with our expectations. But for MART, the trend is not clear as it is not a trade-off based approach.

Table 9: The Correlation between Robust AP/Clean AP and $\lambda$. The results are presented in the format of robust AP/clean AP, with the values representing the average of three runs with different seeds.

| $\lambda$ | 0.01 | 0.4 | 0.8 | 1 | 4 | 8 | 10 |
|---|---|---|---|---|---|---|---|
| CIFAR10(cls_1) | | | | | | | |
| TRADES | 0.0521/0.9609 | 0.4514/0.9135 | 0.4476/0.8846 | 0.4455/0.8727 | 0.4040/0.7930 | 0.3727/0.7479 | 0.3616/0.7305 |
| MART | 0.4511/0.8777 | 0.4548/0.8812 | 0.4519/0.8769 | 0.4349/0.8622 | 0.4247/0.8450 | 0.4016/0.8164 | 0.4030/0.8111 |
| AdAP_LN | 0.1153/0.9525 | 0.4264/0.9295 | 0.4590/0.9190 | 0.4464/0.9099 | 0.3181/0.8649 | 0.2695/0.8385 | 0.2522/0.8356 |
| AdAP_LPN | 0.1504/0.9358 | 0.5185/0.9314 | 0.5263/0.9146 | 0.5229/0.9067 | 0.4818/0.8468 | 0.4517/0.8102 | 0.4411/0.7968 |
| CIFAR100(cls_2) | | | | | | | |
| TRADES | 0.0282/0.8823 | 0.3252/0.7107 | 0.3241/0.6571 | 0.3090/0.6324 | 0.2782/0.5213 | 0.2759/0.5040 | 0.2757/0.4994 |
| MART | 0.3263/0.6669 | 0.3207/0.6538 | 0.3216/0.6527 | 0.3183/0.6420 | 0.3075/0.6095 | 0.2990/0.5839 | 0.2950/0.5767 |
| AdAP_LN | 0.0811/0.8752 | 0.2782/0.7984 | 0.3184/0.7668 | 0.3238/0.7540 | 0.3644/0.6736 | 0.3437/0.6286 | 0.3436/0.6164 |
| AdAP_LPN | 0.1604/0.8332 | 0.3475/0.7896 | 0.3809/0.7558 | 0.3859/0.7441 | 0.3724/0.6470 | 0.3501/0.6011 | 0.3392/0.5827 |
| BDD100K(cloudy) | | | | | | | |
| TRADES | 0.0379/0.6429 | 0.2301/0.5554 | 0.2390/0.5260 | 0.2382/0.5153 | 0.2105/0.4251 | 0.1565/0.2443 | 0.1510/0.2193 |
| MART | 0.2130/0.4459 | 0.2130/0.4450 | 0.2145/0.4441 | 0.2129/0.4404 | 0.1993/0.3973 | 0.1974/0.3901 | 0.1887/0.3631 |
| AdAP_LN | 0.0621/0.6504 | 0.2221/0.5986 | 0.2742/0.5632 | 0.2815/0.5577 | 0.2796/0.4935 | 0.2347/0.4937 | 0.2103/0.4861 |
| AdAP_LPN | 0.0411/0.6570 | 0.2455/0.5985 | 0.2682/0.5681 | 0.2726/0.5600 | 0.2725/0.4886 | 0.2464/0.4366 | 0.2473/0.4295 |
| CelebA(gray_hair) | | | | | | | |
| TRADES | 0.0161/0.7686 | 0.2341/0.5428 | 0.2581/0.4707 | 0.2571/0.4513 | 0.2447/0.3828 | 0.2206/0.3366 | 0.2265/0.3423 |
| MART | 0.2567/0.4910 | 0.2576/0.4882 | 0.2582/0.4865 | 0.2594/0.4864 | 0.2617/0.4784 | 0.2613/0.4696 | 0.2616/0.4672 |
| AdAP_LN | 0.0264/0.7037 | 0.1831/0.6282 | 0.2449/0.6061 | 0.2696/0.6010 | 0.2859/0.5221 | 0.2714/0.4662 | 0.2601/0.4506 |
| AdAP_LPN | 0.0284/0.7123 | 0.2280/0.6259 | 0.2555/0.5907 | 0.2644/0.5804 | 0.2799/0.4948 | 0.2676/0.4515 | 0.2676/0.4515 |

## A.6 Detailed experimental results on CIFAR10 and CIFAR100 Dataset

Due to the limit of space, we include mAP results over 10 classes on CIFAR10 and CIFAR100 datasets in Section 5.1 and Section 5.2. Here, we present the detailed performance for each class on CIFAR10 and CIFAR100 datasets in Table 10, Table 11, Table 12,Table 13, Table 14, Table 15. From Table 10, Table 11, Table 12 and Table 13 about white-box robustness, we can observe that, in most cases, our methods improve other baselines significantly. There is one special case on CIfAR10 Class_3 in Table 10 that MART method behaves much better than others due to high variance(we can see it from the deviation in the parenthesis). When we look at Table 12 about white-box robustness against stronger APGD attacks, we can find that the superiority of MART disappears on CIfAR10 Class_3 since the advantage gained by high variance is not stable. From Table 14 and Table 15 about black-box robustness, we can observe that our methods outperform other baselines by a large margin consistently.

## A.7 More plots for trade-off visualization

Due to page limitations, we present one single trade-off curve for each dataset in Section 5.3 to illustrate our findings. In this part, we include more trade-off curves on all the datasets in Figure 6, Figure 7, Figure 8, Figure 9. These additional figures provide further support for the superiority of our methods.

Table 10: Adversarial robustness against PGD white-box attacks on CIFAR10 dataset

| | CIFAR10 | | | | | | | | | |
|---|---|---|---|---|---|---|---|---|---|---|
| | class_0 | | class_1 | | class_2 | | class_3 | | class_4 | |
| Methods | Robust | Clean | Robust | Clean | Robust | Clean | Robust | Clean | Robust | Clean |
| CE Min. | 0.0519(0.0000) | 0.9037(0.0031) | 0.0558(0.0027) | 0.9687(0.0009) | 0.0518(0.0000) | 0.8493(0.0057) | 0.0518(0.0000) | 0.7235(0.0064) | 0.0519(0.0001) | 0.8843(0.0034) |
| AP Max. | 0.0528(0.0010) | 0.9056(0.0049) | 0.0563(0.0063) | 0.9707(0.0026) | 0.0522(0.0005) | 0.8602(0.0021) | 0.0521(0.0003) | 0.7398(0.0123) | 0.0523(0.0004) | 0.8968(0.0026) |
| PGD | 0.4315(0.0039) | 0.6109(0.0131) | 0.4584(0.0036) | 0.8869(0.0024), | 0.1419(0.0037) | 0.3160(0.0131) | 0.1629(0.0381) | 0.1730(0.0071) | **0.1892(0.0070)** | 0.2523(0.0021) |
| TRADES | 0.4372(0.0032) | 0.6708(0.0033) | 0.4523(0.0049) | 0.9109(0.0011) | 0.1798(0.0060) | 0.3982(0.0094) | 0.1293(0.0004) | 0.2685(0.0029) | 0.1854(0.0009) | 0.3408(0.0028) |
| MART | 0.4295(0.0029) | 0.6032(0.0066) | 0.4559(0.0076) | 0.8796(0.0061) | 0.1528(0.0101) | 0.3068(0.0212) | **0.1942(0.0825)** | 0.1731(0.0143) | 0.1838(0.0025) | 0.2510(0.0014) |
| AdAP_LN | 0.4259(0.0042) | 0.6498(0.0033) | 0.4679(0.0315) | 0.9143(0.0064) | 0.1568(0.0152) | 0.5538(0.0364) | 0.1169(0.0037) | 0.3342(0.0187) | 0.1703(0.0031) | 0.3066(0.0053) |
| AdAP_LPN | **0.4572(0.0024)** | 0.7382(0.0018) | **0.5262(0.0035)** | 0.9145(0.0016) | **0.2041(0.0031)** | 0.5016(0.0047) | 0.1420(0.0008) | 0.2788(0.0038) | 0.1856(0.0007) | 0.3801(0.0062) |
| | class_5 | | class_6 | | class_7 | | class_8 | | class_9 | |
| Methods | Robust | Clean | Robust | Clean | Robust | Clean | Robust | Clean | Robust | Clean |
| CE Min. | 0.0518(0.0000) | 0.8347(0.0050) | 0.0518(0.0000) | 0.9403(0.0048) | 0.0521(0.0002) | 0.9372(0.0022) | 0.0548(0.0031) | 0.9559(0.0022) | 0.0527(0.0005) | 0.9460(0.0031) |
| AP Max. | 0.0518(0.0000) | 0.8420(0.0040) | 0.0519(0.0000) | 0.9419(0.0022) | 0.0519(0.0000) | 0.9416(0.0021) | 0.0545(0.0035) | 0.9569(0.0014) | 0.0518(0.0000) | 0.9513(0.0025) |
| PGD | 0.2329(0.0004) | 0.4524(0.0054) | 0.2363(0.0026) | 0.4111(0.0031) | 0.2356(0.0169) | 0.5928(0.0232) | 0.4157(0.0035) | 0.7390(0.0020) | 0.2602(0.0070) | 0.4944(0.0134) |
| TRADES | 0.2225(0.0017) | 0.4068(0.0047) | 0.2289(0.0016) | 0.3492(0.0029) | 0.3723(0.0022) | 0.7893(0.0011) | 0.3877(0.0054) | 0.7961(0.0159) | 0.2890(0.0028) | 0.7008(0.0012) |
| MART | 0.2348(0.0010) | 0.4510(0.0019) | 0.2346(0.0023) | 0.4059(0.0030) | 0.2917(0.0024) | 0.6667(0.0250) | 0.4158(0.0020) | 0.7397(0.0070) | 0.2662(0.0054) | 0.4935(0.0067) |
| AdAP_LN | 0.2293(0.0067) | 0.5091(0.0063) | 0.2647(0.0197) | 0.7147(0.0106) | 0.3978(0.0258) | 0.8318(0.0075) | 0.4193(0.0287) | 0.8068(0.0019) | 0.3004(0.0084) | 0.7239(0.0176) |
| AdAP_LPN | **0.2467(0.0014)** | 0.5658(0.0028) | **0.2785(0.0019)** | 0.6837(0.0029) | **0.4354(0.0031)** | 0.8292(0.0003) | **0.4564(0.0024)** | 0.8413(0.0028) | **0.3690(0.0030)** | 0.7928(0.0025) |

Table 11: Adversarial robustness against PGD white-box attacks on CIFAR100 dataset

| | class_0 | | class_1 | | class_2 | | class_3 | | class_4 | |
|---|---|---|---|---|---|---|---|---|---|---|
| | CIFAR100 | | | | | | | | | |
| Methods | Robust | Clean | Robust | Clean | Robust | Clean | Robust | Clean | Robust | Clean |
| CE Min. | 0.0256(0.0001) | 0.5465(0.0085) | 0.0256(0.0001) | 0.6112(0.0025) | 0.0255(0.0000) | 0.8932(0.0025) | 0.0255(0.0000) | 0.7359(0.0072) | 0.0258(0.0000) | 0.8255(0.0078) |
| AP Max. | 0.0255(0.0000) | 0.5480(0.0060) | 0.0258(0.0002) | 0.6348(0.0199) | 0.0259(0.0003) | 0.8993(0.0022) | 0.0255(0.0000) | 0.7392(0.0079) | 0.0267(0.0015) | 0.8355(0.0035) |
| PGD | 0.1124(0.0021) | 0.2128(0.0060) | 0.1845(0.0020) | 0.3414(0.0071) | 0.3261(0.0066) | 0.6618(0.0036) | 0.1141(0.0066) | 0.2466(0.0120) | 0.3115(0.0022) | 0.5166(0.0034) |
| TRADES | 0.1177(0.0010) | 0.2560(0.0020) | 0.1942(0.0028) | 0.3888(0.0069) | 0.3289(0.0044) | 0.6937(0.0106) | 0.1663(0.0053) | 0.3976(0.0119) | 0.3039(0.0031) | 0.5289(0.0060) |
| MART | 0.1187(0.0027) | 0.2188(0.0029) | 0.1882(0.0037) | 0.3417(0.0080) | 0.3258(0.0074) | 0.6651(0.0034) | 0.1352(0.0144) | 0.2389(0.0092) | 0.3148(0.0043) | 0.5140(0.0061) |
| AdAP_LN | 0.1273(0.0028) | 0.3131(0.0049) | 0.1912(0.0020) | 0.4073(0.0036) | 0.3658(0.0054) | 0.6736(0.0035) | 0.1930(0.0011) | 0.4403(0.0042) | 0.3396(0.0174) | 0.5580(0.0036) |
| AdAP_LPN | **0.1376(0.0005)** | 0.2946(0.0014) | **0.2144(0.0069)** | 0.4469(0.0092) | **0.3916(0.0096)** | 0.7342(0.0111) | **0.2085(0.0045)** | 0.4979(0.0019) | **0.3554(0.0005)** | 0.5439(0.0014) |

| | class_5 | | class_6 | | class_7 | | class_8 | | class_9 | |
|---|---|---|---|---|---|---|---|---|---|---|
| Methods | Robust | Clean | Robust | Clean | Robust | Clean | Robust | Clean | Robust | Clean |
| CE Min. | 0.0255(0.0000) | 0.5990(0.0065) | 0.0255(0.0000) | 0.7690(0.0039) | 0.0255(0.0000) | 0.6743(0.0037) | 0.0255(0.0000) | 0.6033(0.0102) | 0.0262(0.0009) | 0.8361(0.0095) |
| AP Max. | 0.0299(0.0054) | 0.6257(0.0081) | 0.0262(0.0005) | 0.7806(0.0032) | 0.0276(0.0029) | 0.6974(0.0069) | 0.0255(0.0000) | 0.6192(0.0111) | 0.0282(0.0038) | 0.8422(0.0082) |
| PGD | 0.0804(0.0025) | 0.1671(0.0151) | 0.1054(0.0018) | 0.1960(0.0090) | 0.1441(0.0021) | 0.3205(0.0057) | 0.0763(0.0108) | 0.0943(0.0066) | 0.1583(0.0060) | 0.3915(0.0147) |
| TRADES | 0.0963(0.0036) | 0.2972(0.0068) | 0.1533(0.0031) | 0.4387(0.0059) | 0.1751(0.0011) | 0.3660(0.0008) | 0.0662(0.0008) | 0.1174(0.0054) | 0.1655(0.0013) | 0.4910(0.0086) |
| MART | 0.0805(0.0028) | 0.1439(0.0060) | 0.1095(0.0079) | 0.2531(0.0100) | 0.1477(0.0020) | 0.2973(0.0030) | 0.0700(0.0100) | 0.0969(0.0065) | 0.1593(0.0047) | 0.3972(0.0157) |
| AdAP_LN | 0.1131(0.0040) | 0.3900(0.0524) | 0.2058(0.0060) | 0.5916(0.0130) | 0.1884(0.0020) | 0.4611(0.0076) | 0.0693(0.0119) | 0.2031(0.0334) | 0.1631(0.0160) | 0.5897(0.0117) |
| AdAP_LPN | **0.1483(0.0038)** | 0.4268(0.0164) | **0.2459(0.0021)** | 0.5670(0.0153) | **0.2170(0.0037)** | 0.4234(0.0062) | 0.0710(0.0003) | 0.1552(0.0018) | **0.1986(0.0083)** | 0.5846(0.0105) |

Table 12: Adversarial robustness against APGD white-box attacks on CIFAR10 dataset

| | class_0 | | class_1 | | class_2 | | class_3 | | class_4 | |
|---|---|---|---|---|---|---|---|---|---|---|
| | CIFAR10 | | | | | | | | | |
| Methods | Robust | Clean | Robust | Clean | Robust | Clean | Robust | Clean | Robust | Clean |
| CE Min. | 0.0518(0.0000) | 0.9037(0.0031) | 0.0547(0.0017) | 0.9687(0.0009) | 0.0518(0.0000) | 0.8493(0.0057) | 0.0518(0.0000) | 0.7235(0.0064) | 0.0519(0.0001) | 0.8843(0.0034) |
| AP Max. | 0.0519(0.0001) | 0.9056(0.0049) | 0.0596(0.0110) | 0.9707(0.0026) | 0.0518(0.0000) | 0.8602(0.0021) | 0.0518(0.0000) | 0.7398(0.0123) | 0.0519(0.0000) | 0.8968(0.0034) |
| PGD | 0.4291(0.0038) | 0.6109(0.0131) | 0.4482(0.0033) | 0.8869(0.0024,) | 0.1380(0.0034) | 0.3160(0.0131) | 0.1120(0.0145) | 0.1730(0.0071) | 0.1797(0.0014) | 0.2523(0.0021) |
| TRADES | 0.4334(0.0031) | 0.6708(0.0033) | 0.4391(0.0050) | 0.9109(0.0011) | 0.1782(0.0062) | 0.3982(0.0094) | 0.1289(0.0044) | 0.2685(0.0029) | **0.1847(0.0046)** | 0.3408(0.0028) |
| MART | 0.4274(0.0032) | 0.6032(0.0066) | 0.4465(0.0074) | 0.8796(0.0061) | 0.1509(0.0105) | 0.3068(0.0212) | 0.0929(0.0092) | 0.1731(0.0143) | 0.1756(0.0019) | 0.2510(0.0014) |
| AdAP_LN | 0.4222(0.0034) | 0.6498(0.0033) | 0.4220(0.0117) | 0.9143(0.0064) | 0.1371(0.0074) | 0.5538(0.0364) | 0.1144(0.0024) | 0.3342(0.0187) | 0.1701(0.0033) | 0.3066(0.0053) |
| AdAP_LPN | **0.4529(0.0025)** | 0.7382(0.0018) | **0.5153(0.0031)** | 0.9145(0.0016) | **0.2012(0.0030)** | 0.5016(0.0047) | **0.1413(0.0008)** | 0.2788(0.0038) | 0.1843(0.0008) | 0.3801(0.0062) |

| | class_5 | | class_6 | | class_7 | | class_8 | | class_9 | |
|---|---|---|---|---|---|---|---|---|---|---|
| Methods | Robust | Clean | Robust | Clean | Robust | Clean | Robust | Clean | Robust | Clean |
| CE Min. | 0.0518(0.0000) | 0.8347(0.0050) | 0.0518(0.0000) | 0.9403(0.0048) | 0.0520(0.0001) | 0.9372(0.0022) | 0.0537(0.0019) | 0.9559(0.0022) | 0.0525(0.0004) | 0.9460(0.0031) |
| AP Max. | 0.0518(0.0000) | 0.8420(0.0040) | 0.0519(0.0002) | 0.9419(0.0022) | 0.0518(0.0000) | 0.9416(0.0021) | 0.0519(0.0002) | 0.9569(0.0014) | 0.0518(0.0000) | 0.9513(0.0025) |
| PGD | 0.2314(0.0003) | 0.4524(0.0054) | 0.2350(0.0028) | 0.4111(0.0031) | 0.2270(0.0145) | 0.5928(0.0232) | 0.4113(0.0034) | 0.7390(0.0020) | 0.2580(0.0066) | 0.4944(0.0134) |
| TRADES | 0.2217(0.0017) | 0.4068(0.0047) | 0.2284(0.0016) | 0.3492(0.0029) | 0.3627(0.0019) | 0.7893(0.0011) | 0.3790(0.0057) | 0.7961(0.0159) | 0.2829(0.0030) | 0.7008(0.0012) |
| MART | 0.2333(0.0009) | 0.4510(0.0019) | 0.2334(0.0023) | 0.4059(0.0030) | 0.2853(0.0094) | 0.6667(0.0250) | 0.4117(0.0017) | 0.7397(0.0070) | 0.2642(0.0055) | 0.4935(0.0067) |
| AdAP_LN | 0.2209(0.0030) | 0.5091(0.0063) | 0.1555(0.0096) | 0.7147(0.0106) | 0.3591(0.0060) | 0.8318(0.0075) | 0.3434(0.0055) | 0.8068(0.0019) | 0.2488(0.0041) | 0.7239(0.0176) |
| AdAP_LPN | **0.2436(0.0015)** | 0.5658(0.0028) | **0.2737(0.0014)** | 0.6837(0.0029) | **0.4276(0.0033)** | 0.8292(0.0003) | **0.4496(0.0025)** | 0.8413(0.0028) | **0.3623(0.0029)** | 0.7928(0.0025) |

Table 13: Adversarial robustness against APGD white-box attack on CIFAR100 dataset

| | class_0 | | class_1 | | class_2 | | class_3 | | class_4 | |
|---|---|---|---|---|---|---|---|---|---|---|
| | CIFAR100 | | | | | | | | | |
| Methods | Robust | Clean | Robust | Clean | Robust | Clean | Robust | Clean | Robust | Clean |
| CE Min. | 0.0255(0.0000) | 0.5465(0.0085) | 0.0255(0.0000) | 0.6112(0.0025) | 0.0255(0.0000) | 0.8932(0.0025) | 0.0255(0.0000) | 0.7359(0.0072) | 0.0257(0.0000) | 0.8255(0.0078) |
| AP Max. | 0.0255(0.0000) | 0.5480(0.0060) | 0.0255(0.0000) | 0.6348(0.0199) | 0.0255(0.0000) | 0.8993(0.0022) | 0.0255(0.0000) | 0.7392(0.0079) | 0.0264(0.0013) | 0.8355(0.0035) |
| PGD | 0.1103(0.0017) | 0.2128(0.0060) | 0.1831(0.0020) | 0.3414(0.0071) | 0.3232(0.0063) | 0.6618(0.0036) | 0.1100(0.0058) | 0.2466(0.0120) | 0.3097(0.0021) | 0.5166(0.0034) |
| TRADES | 0.1171(0.0011) | 0.2560(0.0020) | 0.1929(0.0028) | 0.3888(0.0069) | 0.3224(0.0048) | 0.6937(0.0106) | 0.1632(0.0051) | 0.3976(0.0119) | 0.3020(0.0032) | 0.5289(0.0060) |
| MART | 0.1101(0.0046) | 0.2188(0.0029) | 0.1864(0.0033) | 0.3417(0.0080) | 0.3229(0.0073) | 0.6651(0.0034) | 0.1279(0.0112) | 0.2389(0.0092) | 0.3129(0.0043) | 0.5140(0.0061) |
| AdAP_LN | 0.1245(0.0024) | 0.3131(0.0049) | 0.1896(0.0020) | 0.4073(0.0036) | 0.3566(0.0042) | 0.6736(0.0035) | 0.1878(0.0011) | 0.4403(0.0042) | 0.3235(0.0048) | 0.5580(0.0036) |
| AdAP_LPN | **0.1367(0.0007)** | 0.2946(0.0014) | **0.2128(0.0068)** | 0.4469(0.0092) | **0.3871(0.0093)** | 0.7342(0.0111) | **0.2039(0.0047)** | 0.4979(0.0019) | **0.3535(0.0005)** | 0.5439(0.0014) |

| | class_5 | | class_6 | | class_7 | | class_8 | | class_9 | |
|---|---|---|---|---|---|---|---|---|---|---|
| Methods | Robust | Clean | Robust | Clean | Robust | Clean | Robust | Clean | Robust | Clean |
| CE Min. | 0.0255(0.0000) | 0.5990(0.0065) | 0.0255(0.0000) | 0.7690(0.0039) | 0.0255(0.0000) | 0.6743(0.0037) | 0.0255(0.0000) | 0.6033(0.0102) | 0.0255(0.0000) | 0.8361(0.0095) |
| AP Max. | 0.0255(0.0000) | 0.6257(0.0081) | 0.0255(0.0000) | 0.7806(0.0032) | 0.0264(0.0013) | 0.6974(0.0069) | 0.0255(0.0000) | 0.6192(0.0111) | 0.0256(0.0001) | 0.8422(0.0082) |
| PGD | 0.0777(0.0033) | 0.1671(0.0151) | 0.1047(0.0019) | 0.1960(0.0090) | 0.1418(0.0025) | 0.3205(0.0057) | 0.0632(0.0035) | 0.0943(0.0066) | 0.1561(0.0059) | 0.3915(0.0147) |
| TRADES | 0.0943(0.0033) | 0.2972(0.0068) | 0.1438(0.0036) | 0.4387(0.0059) | 0.1741(0.0011) | 0.3660(0.0008) | 0.0660(0.0008) | 0.1174(0.0054) | 0.1633(0.0014) | 0.4910(0.0086) |
| MART | 0.0789(0.0040) | 0.1439(0.0060) | 0.1037(0.0081) | 0.2531(0.0100) | 0.1458(0.0026) | 0.2973(0.0030) | 0.0575(0.0040) | 0.0969(0.0065) | 0.1572(0.0047) | 0.3972(0.0157) |
| AdAP_LN | 0.1085(0.0027) | 0.3900(0.0524) | 0.1871(0.0074) | 0.5916(0.0130) | 0.1840(0.0014) | 0.4611(0.0076) | 0.0558(0.0009) | 0.2031(0.0334) | 0.1352(0.0012) | 0.5897(0.0117) |
| AdAP_LPN | **0.1444(0.0035)** | 0.4268(0.0164) | **0.2421(0.0017)** | 0.5670(0.0153) | **0.2150(0.0041)** | 0.4234(0.0062) | **0.0706(0.0003)** | 0.1552(0.0018) | **0.1943(0.0084)** | 0.5846(0.0105) |

Table 14: Adversarial robustness against black-box attack on CIFAR10 dataset

| | class_0 | | class_1 | | class_2 | | class_3 | | class_4 | |
|---|---|---|---|---|---|---|---|---|---|---|
| | CIFAR10 | | | | | | | | | |
| Methods | Robust | Clean | Robust | Clean | Robust | Clean | Robust | Clean | Robust | Clean |
| AP Max. | 0.1038(0.0362) | 0.9056(0.0049) | 0.2553(0.1270) | 0.9707(0.0026) | 0.0614(0.0063) | 0.8602(0.0021) | 0.0634(0.0069) | 0.7398(0.0123) | 0.0989(0.0330) | 0.8968(0.0026) |
| PGD | 0.5773(0.0113) | 0.6109(0.0131) | 0.8229(0.0031) | 0.8869(0.0024,) | 0.3014(0.0125) | 0.3160(0.0131) | 0.1629(0.0057) | 0.1730(0.0071) | 0.2438(0.0023) | 0.2523(0.0021) |
| TRADES | 0.6198(0.0030) | 0.6708(0.0033) | 0.8430(0.0003) | 0.9109(0.0011) | 0.3597(0.0074) | 0.3982(0.0094) | 0.2483(0.0028) | 0.2685(0.0029) | 0.3244(0.0029) | 0.3408(0.0028) |
| MART | 0.5699(0.0060) | 0.6032(0.0066) | 0.8147(0.0054) | 0.8796(0.0061) | 0.2886(0.0177) | 0.3068(0.0212) | 0.1629(0.0105) | 0.1731(0.0143) | 0.2455(0.0023) | 0.2510(0.0014) |
| AdAP_LN | 0.6079(0.0034) | 0.6498(0.0033) | 0.8444(0.0037) | 0.9143(0.0064) | 0.4365(0.0254) | 0.5538(0.0364) | **0.2873(0.0152)** | 0.3342(0.0187) | 0.2913(0.0039) | 0.3066(0.0053) |
| AdAP_LPN | **0.6888(0.0018)** | 0.7382(0.0018) | **0.8650(0.0011)** | 0.9145(0.0016) | **0.4503(0.0057)** | 0.5016(0.0047) | 0.2550(0.0029) | 0.2788(0.0039) | **0.3562(0.0057)** | 0.3801(0.0062) |

| | class_5 | | class_6 | | class_7 | | class_8 | | class_9 | |
|---|---|---|---|---|---|---|---|---|---|---|
| Methods | Robust | Clean | Robust | Clean | Robust | Clean | Robust | Clean | Robust | Clean |
| AP Max. | 0.0723(0.0128) | 0.8420(0.0040) | 0.0947(0.0245) | 0.9419(0.0022) | 0.1008(0.0342) | 0.9416(0.0021) | 0.1153(0.0355) | 0.9569(0.0014) | 0.1712(0.0578) | 0.9513(0.0025) |
| PGD | 0.4204(0.0042) | 0.4524(0.0054) | 0.3973(0.0035) | 0.4111(0.0031) | 0.5522(0.0198) | 0.5928(0.0232) | 0.6732(0.0005) | 0.7390(0.0020) | 0.4427(0.0112) | 0.4944(0.0134) |
| TRADES | 0.3816(0.0036) | 0.4068(0.0047) | 0.3358(0.0028) | 0.3492(0.0029) | 0.7100(0.0019) | 0.7893(0.0011) | 0.7034(0.0127) | 0.7961(0.0159) | 0.6216(0.0012) | 0.7008(0.0012) |
| MART | 0.4199(0.0015) | 0.4510(0.0019) | 0.3924(0.0025) | 0.4059(0.0030) | 0.6145(0.0249) | 0.6667(0.0250) | 0.6727(0.0071) | 0.7397(0.0070) | 0.4428(0.0050) | 0.4935(0.0067) |
| AdAP_LN | 0.4582(0.0042) | 0.5091(0.0063) | 0.5564(0.0016) | 0.7147(0.0106) | 0.7371(0.0049) | 0.8318(0.0075) | 0.7107(0.0022) | 0.8068(0.0019) | 0.6322(0.0224) | 0.7239(0.0176) |
| AdAP_LPN | **0.5200(0.0024)** | 0.5658(0.0028) | **0.6277(0.0029)** | 0.6837(0.0028) | **0.7652(0.0003)** | 0.8292(0.0003) | **0.7689(0.0017)** | 0.8413(0.0029) | **0.7338(0.0031)** | 0.7928(0.0025) |

Table 15: Adversarial robustness against black-box attack on CIFAR100 dataset

| | class_0 | | class_1 | | class_2 | | class_3 | | class_4 | |
|---|---|---|---|---|---|---|---|---|---|---|
| | CIFAR100 | | | | | | | | | |
| Methods | Robust | Clean | Robust | Clean | Robust | Clean | Robust | Clean | Robust | Clean |
| AP Max. | 0.0324(0.0046) | 0.5480(0.0060) | 0.0632(0.0267) | 0.6348(0.0199) | 0.1081(0.0581) | 0.8993(0.0022) | 0.0411(0.0111) | 0.7392(0.0079) | 0.1034(0.0547) | 0.8355(0.0035) |
| PGD | 0.1816(0.0051) | 0.2128(0.0060) | 0.3136(0.0068) | 0.3414(0.0071) | 0.6056(0.0058) | 0.6618(0.0036) | 0.2366(0.0119) | 0.2466(0.0120) | 0.4816(0.0025) | 0.5166(0.0034) |
| TRADES | 0.2184(0.0013) | 0.2560(0.0020) | 0.3454(0.0031) | 0.3888(0.0069) | 0.6138(0.0088) | 0.6937(0.0106) | 0.3538(0.0095) | 0.3976(0.0119) | 0.4858(0.0030) | 0.5289(0.0060) |
| MART | 0.1869(0.0042) | 0.2188(0.0029) | 0.3145(0.0079) | 0.3417(0.0080) | 0.6093(0.0038) | 0.6651(0.0034) | 0.2260(0.0079) | 0.2389(0.0092) | 0.4812(0.0055) | 0.5140(0.0061) |
| AdAP_LN | 0.2534(0.0022) | 0.3131(0.0049) | 0.3547(0.0038) | 0.4073(0.0036) | 0.6101(0.0032) | 0.6736(0.0035) | 0.3961(0.0053) | 0.4403(0.0042) | **0.5139(0.0038)** | 0.5580(0.0036) |
| AdAP_LPN | **0.2548(0.0015)** | 0.2946(0.0014) | **0.3983(0.0075)** | 0.4469(0.0092) | **0.6703(0.0106)** | 0.7342(0.0111) | **0.4542(0.0021)** | 0.4979(0.0020) | 0.5136(0.0024) | 0.5439(0.0014) |

| | class_5 | | class_6 | | class_7 | | class_8 | | class_9 | |
|---|---|---|---|---|---|---|---|---|---|---|
| Methods | Robust | Clean | Robust | Clean | Robust | Clean | Robust | Clean | Robust | Clean |
| AP Max. | 0.0820(0.0376) | 0.6257(0.0081) | 0.0355(0.0079) | 0.7806(0.0032) | 0.0593(0.0231) | 0.6974(0.0069) | 0.0360(0.0064) | 0.6192(0.0111) | 0.0470(0.0112) | 0.8422(0.0082) |
| PGD | 0.1532(0.0143) | 0.1671(0.0151) | 0.1692(0.0053) | 0.1960(0.0090) | 0.2976(0.0059) | 0.3205(0.0057) | 0.0901(0.0051) | 0.0943(0.0066) | 0.3403(0.0149) | 0.3915(0.0147) |
| TRADES | 0.2389(0.0062) | 0.2972(0.0068) | 0.3338(0.0036) | 0.4387(0.0059) | 0.3290(0.0003) | 0.3660(0.0008) | 0.1021(0.0024) | 0.1174(0.0054) | 0.3994(0.0071) | 0.4910(0.0086) |
| MART | 0.1256(0.0055) | 0.1439(0.0060) | 0.2304(0.0102) | 0.2531(0.0100) | 0.2770(0.0025) | 0.2973(0.0030) | 0.0938(0.0059) | 0.0969(0.0065) | 0.3475(0.0133) | 0.3972(0.0157) |
| AdAP_LN | 0.3113(0.0374) | 0.3900(0.0524) | 0.4656(0.0114) | 0.5916(0.0130) | **0.3969(0.0047)** | 0.4611(0.0076) | **0.1338(0.0094)** | 0.2031(0.0334) | 0.4442(0.0179) | 0.5897(0.0117) |
| AdAP_LPN | **0.3701(0.0149)** | 0.4268(0.0164) | **0.4929(0.0140)** | 0.5670(0.0153) | 0.3867(0.0058) | 0.4234(0.0062) | 0.1302(0.0013) | 0.1552(0.0018) | **0.4948(0.0116)** | 0.5846(0.0105) |

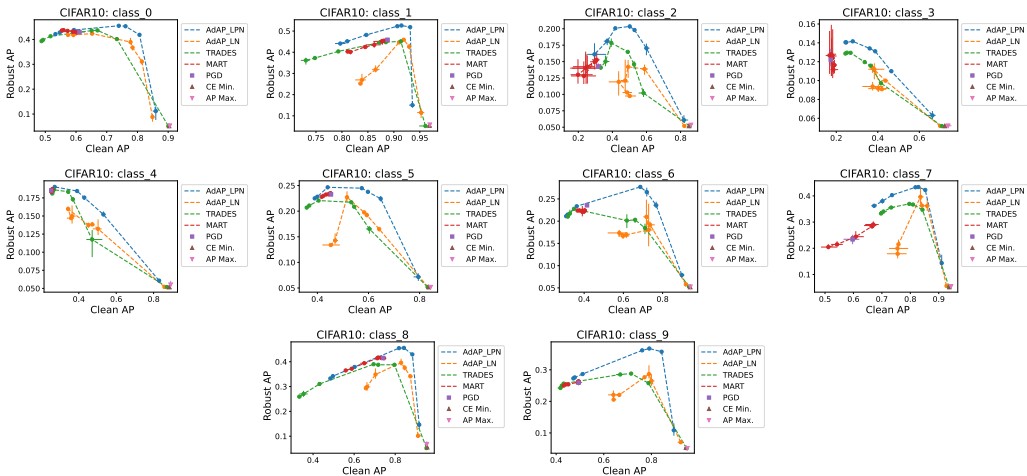

Figure 6: Visualization of the trade-off between robustness and average precision on CIFAR10 dataset

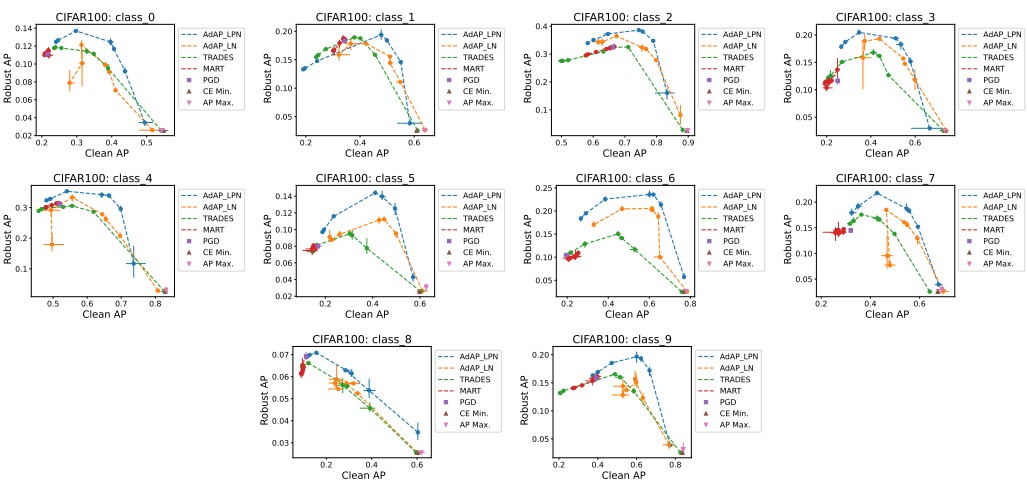

Figure 7: Visualization of the trade-off between robustness and average precision on CIFAR100 dataset

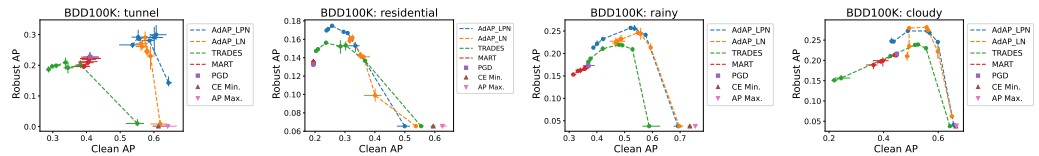

Figure 8: Visualization of the trade-off between robustness and average precision on BDD100K dataset

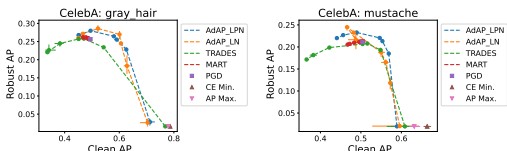

Figure 9: Visualization of the trade-off between robustness and average precision on CelebA dataset

# B  Theoretical Results

## B.1  Proof of theorem 1

Note that robust ranking error is defined as:
$$\mathcal{R}_{rob} = \mathbb{E}_{\mathbf{x}_i \sim \mathcal{D}_+} \mathbb{I}\{\exists \delta_i, \delta_j \in \mathbb{B}(0, \epsilon), h(\mathbf{x}_i + \delta_i) \leq \max_{\mathbf{x}_j \in \mathcal{D}_-} h(\mathbf{x}_j + \delta_j)\}. \tag{13}$$

Then we can reformulate it as:
$$\mathcal{R}_{rob} = \Pr\left(\exists \delta_i, \delta_j \in \mathbb{B}(0, \epsilon), h(\mathbf{x}_i + \delta_i) \leq \max_{\mathbf{x}_j \in \mathcal{D}_-} h(\mathbf{x}_j + \delta_j) \mid \mathbf{x}_i \in \mathcal{D}_+\right)$$

$$= \Pr\left(\exists \delta_i, \delta_j \in \mathbb{B}(0, \epsilon), h(\mathbf{x}_i + \delta_i) \leq \max_{\mathbf{x}_j \in \mathcal{D}_-} h(\mathbf{x}_j + \delta_j) \mid h(\mathbf{x}_i) > \max_{\mathbf{x}_j \in \mathcal{D}_-} h(\mathbf{x}_j), \mathbf{x}_i \in \mathcal{D}_+\right)$$

$$* \Pr\left(h(\mathbf{x}_i) > \max_{\mathbf{x}_j \in \mathcal{D}_-} h(\mathbf{x}_j) \mid \mathbf{x}_i \in \mathcal{D}_+\right)$$

$$+ \Pr\left(\exists \delta_i, \delta_j \in \mathbb{B}(0, \epsilon), h(\mathbf{x}_i + \delta_i) \leq \max_{\mathbf{x}_j \in \mathcal{D}_-} h(\mathbf{x}_j + \delta_j) \mid h(\mathbf{x}_i) \leq \max_{\mathbf{x}_j \in \mathcal{D}_-} h(\mathbf{x}_j), \mathbf{x}_i \in \mathcal{D}_+\right)$$

$$* \Pr\left(h(\mathbf{x}_i) \leq \max_{\mathbf{x}_j \in \mathcal{D}_-} h(\mathbf{x}_j) \mid \mathbf{x}_i \in \mathcal{D}_+\right)$$

$$= \Pr\left(\{(h(\mathbf{x}_i) > \max_{\mathbf{x}_j \in \mathcal{D}_-} h(\mathbf{x}_j)\} \wedge \{\exists \delta_i, \delta_j \in \mathbb{B}(0, \epsilon), h(\mathbf{x}_i + \delta_i) \leq \max_{\mathbf{x}_j \in \mathcal{D}_-} h(\mathbf{x}_j + \delta_j)\} \mid \mathbf{x}_i \in \mathcal{D}_+)\right)$$

$$+ 1 * \Pr\left(h(\mathbf{x}_i) \leq \max_{\mathbf{x}_j \in \mathcal{D}_-} h(\mathbf{x}_j) \mid \mathbf{x}_i \in \mathcal{D}_+\right)$$

$$= \mathbb{E}_{\mathbf{x}_i \sim \mathcal{D}_+} \mathbb{I}\{h(\mathbf{x}_i) > \max_{\mathbf{x}_j \in \mathcal{D}_-} h(\mathbf{x}_j)\} \mathbb{I}\{\exists \delta_i, \delta_j \in \mathbb{B}(0, \epsilon), h(\mathbf{x}_i + \delta_i) \leq \max_{\mathbf{x}_j \in \mathcal{D}_-} h(\mathbf{x}_j + \delta_j)\}$$

$$+ \mathbb{E}_{\mathbf{x}_i \sim \mathcal{D}_+} \mathbb{I}\{h(\mathbf{x}_i) \leq \max_{\mathbf{x}_j \in \mathcal{D}_-} h(\mathbf{x}_j)\}$$

Notice that the last two terms above are exactly the considered boundary error $\mathcal{R}_{bdy}$ and misranking error $\mathcal{R}_{nat}$. Therefore, we have $\mathcal{R}_{rob} = \mathcal{R}_{bdy} + \mathcal{R}_{nat}$.

## B.2  Extra algorithms for adversarial AP maximization

Besides the Algorithm 1 shown in Section 4.2, we present the other algorithms for adversarial AP maximization in this part. Algorithms for solving AdAP_PZ in (7) and AdAP_LPN in (10) are similar to the Algorithm 1 and we present them in Algorithm 4 and Algorithm 3.

$$\min_{\mathbf{w}} \max_{\|\delta\| \leq \epsilon} P(\mathbf{w}, \delta) = -\frac{1}{n_+} \sum_{\mathbf{x}_i \in \mathcal{D}_+} \frac{\sum_{s=1}^{n} \mathbb{I}(y_s = 1) \ell(\mathbf{w}, \mathbf{x}_s + \delta_s, \mathbf{x}_i + \delta_i)}{\sum_{s=1}^{n} \ell(\mathbf{w}, \mathbf{x}_s + \delta_s, \mathbf{x}_i + \delta_i)}. \tag{14}$$

Different from above algorithms, the algorithm for solving AdAP_MM in (14) needs to solve inner maximization with AP surrogate loss to generate adversarial perturbations. Because this inner maximization problem includes all the samples in the dataset, we apply the similar compositional optimization technique borrowed from [34] to compute an estimate of $\nabla_\delta P(\mathbf{w}, \delta)$ by:

$$\widehat{\nabla}_\delta P(\mathbf{w}, \delta) = \frac{1}{|\mathcal{B}^+|} \sum_{\mathbf{x}_i \in \mathcal{B}^+} \sum_{\mathbf{x}_j \in \mathcal{B}} \frac{(\mathbf{u}_{\mathbf{x}_i}^{\delta 1} - \mathbf{u}_{\mathbf{x}_i}^{\delta 2} \mathbb{I}(\mathbf{y}_j = 1)) \nabla_\delta \ell(\mathbf{w}, \mathbf{x}_j + \delta_j, \mathbf{x}_i + \delta_i)}{|\mathcal{B}|(\mathbf{u}_{\mathbf{x}_i}^{\delta 2})^2}. \tag{15}$$

where $\mathbf{u}_{\mathbf{x}_i}^{\delta 1}$ and $\mathbf{u}_{\mathbf{x}_i}^{\delta 2}$ are two estimators for tracking $\varphi_{\delta 1}(\mathbf{w}, \delta_i; \mathbf{x}_i) = \frac{1}{n} \sum_{s=1}^{n} \mathbb{I}(y_s = 1) \ell(\mathbf{w}, \mathbf{x}_s + \delta_s, \mathbf{x}_i + \delta_i)$, $\varphi_{\delta 2}(\mathbf{w}, \delta_i; \mathbf{x}_i) = \frac{1}{n} \sum_{s=1}^{n} \ell(\mathbf{w}, \mathbf{x}_s + \delta_s, \mathbf{x}_i + \delta_i)$. At iteration $t$, the two estimators $\mathbf{u}_{\mathbf{x}_i}^{\delta 1}$ and $\mathbf{u}_{\mathbf{x}_i}^{\delta 2}$ are updated by

$$\mathbf{u}_{\mathbf{x}_i}^{\delta 1} = (1 - \gamma_1) \mathbf{u}_{\mathbf{x}_i}^{\delta 1} + \gamma_1 \frac{1}{|\mathcal{B}|} \sum \ell(\mathbf{w}_t, \mathbf{x}_s + \delta_s, \mathbf{x}_i + \delta_i) \mathbb{I}(y_j = 1)$$

$$\mathbf{u}_{\mathbf{x}_i}^{\delta 2} = (1 - \gamma_1) \mathbf{u}_{\mathbf{x}_i}^{\delta 2} + \gamma_1 \frac{1}{|\mathcal{B}|} \sum \ell(\mathbf{w}_t, \mathbf{x}_s + \delta_s, \mathbf{x}_i + \delta_i), \tag{16}$$

where $\gamma_1 \in (0, 1)$ is a parameter. And the stochastic gradient $\widehat{\nabla}_{\mathbf{w}} P(\mathbf{w}, \delta)$ is computed by:

$$\widehat{\nabla}_{\mathbf{w}} P(\mathbf{w}, \delta) = \frac{1}{|\mathcal{B}^+|} \sum_{\mathbf{x}_i \in \mathcal{B}^+} \sum_{\mathbf{x}_j \in \mathcal{B}} \frac{(\mathbf{u}_{\mathbf{x}_i}^{\delta 1} - \mathbf{u}_{\mathbf{x}_i}^{\delta 2} \mathbb{I}(\mathbf{y}_j = 1)) \nabla_{\mathbf{w}} \ell(\mathbf{w}, \mathbf{x}_j + \delta_j, \mathbf{x}_i + \delta_i)}{|\mathcal{B}|(\mathbf{u}_{\mathbf{x}_i}^{\delta 2})^2}. \quad (17)$$

We present the whole algorithm for solving AdAP_MM in (14) in Algorithm 2.

The objective for AdAP_LZ is defined as

$$\min_{\mathbf{w}} \max_{\delta \in \Omega^n} P(\mathbf{w}) + R(\mathbf{w}, \delta, \mathcal{D}) = \min_{\mathbf{w}} \left\{ P(\mathbf{w}) + \max_{\delta \in \Omega^n} R(\mathbf{w}, \delta, \mathcal{D}) \right\} \quad (18)$$

Since the listwise adversarial regularization includes all the samples in the dataset, the optimization for $\delta$ is also challenging. We extend the approach for computing an estimator of $\nabla_{\mathbf{w}} R(\mathbf{w}, \delta, \mathcal{D})$ employed in Section 4.2 to estimate $\nabla_\delta R(\mathbf{w}, \delta, \mathcal{D})$. To avoid redundancy, we give the estimate directly by

$$\widehat{\nabla}_\delta R(\mathbf{w}, \delta, \mathcal{D}) = \widehat{\nabla}_\delta g(\delta; \mathbf{w})^\top \left( \frac{1}{u_2}, \frac{-u_1 - u_2}{(u_2)^2}, \frac{1}{u_3} \right)^\top, \quad (19)$$

where $g(\delta; \mathbf{w})$ has the same definition as $g(\mathbf{w})$ in Section 4.2, $\mathbf{u} = [u_1, u_2, u_3]$ denotes a vector for tracking $[g_1(\mathbf{w}), g_2(\mathbf{w}), g_3(\mathbf{w})]$. The whole algorithm for solving AdAP_LZ is presented in Algorithm 5.

---

**Algorithm 2** Stochastic Algorithm for solving AdAP_MM in (14)

---

1: Initialize $\mathbf{w}, \mathbf{u}_{\mathbf{x}}^{\delta 1}, \mathbf{u}_{\mathbf{x}}^{\delta 2}, \gamma_1$
2: **for** $t = 1, \ldots, T$ **do**
3:  Draw a batch of $B_+$ positive samples denoted by $\mathcal{B}_+$.
4:  Draw a batch of $B$ samples denoted by $\mathcal{B}$.
5:  For $\mathbf{x}_i \in \mathcal{B} \cup \mathcal{B}_+$, initialize $\delta_i \sim \alpha \cdot \mathcal{N}(0, 1)$
6:  **for** $m = 1, \ldots, M$ **do**
7:    For each $\mathbf{x}_i \in \mathcal{B}_+$, update $\mathbf{u}_{\mathbf{x}_i}^{\delta 1}$ and $\mathbf{u}_{\mathbf{x}_i}^{\delta 2}$ by Equation (16)
8:    Compute $\widehat{\nabla}_\delta P(\mathbf{w}_t, \delta)$ by Equation (15)
9:    Update $\delta_i = \Pi_{\|\cdot\| \le \epsilon}(\delta_i + \eta_2 \cdot sign(\widehat{\nabla}_{\delta_i} P(\mathbf{w}_t, \delta)))$, where $\Pi_\Omega(\cdot)$ is the projection operator.
10:  **end for**
11:  For each $\mathbf{x}_i \in \mathcal{B}_+$, update $\mathbf{u}_{\mathbf{x}_i}^{\delta 1}$ and $\mathbf{u}_{\mathbf{x}_i}^{\delta 2}$ by Equation (16)
12:  Compute stochastic gradient estimator $\nabla_{\mathbf{w}_t} = \widehat{\nabla}_{\mathbf{w}_t} P(\mathbf{w}_t, \delta)$ by Equation (17)
13:  Update $\mathbf{w}_{t+1}$ by using SGD, momentum-methods or Adam.
14: **end for**

---

**Algorithm 3** Stochastic Algorithm for solving AdAP_LPN in (10)

---

1: Initialize $\mathbf{w}, \mathbf{u}_{\mathbf{x}}, \mathbf{u}, \gamma_1, \gamma_2$
2: **for** $t = 1, \ldots, T$ **do**
3:  Draw a batch of $B_+$ positive samples denoted by $\mathcal{B}_+$.
4:  Draw a batch of $B$ samples denoted by $\mathcal{B}$.
5:  **for** $\mathbf{x}_i \in \mathcal{B}$ **do**
6:    Initialize $\delta_i \sim \alpha \cdot \mathcal{N}(0, 1)$
7:    **for** $m = 1, \ldots, M$ **do**
8:      Update $\delta_i = \Pi_{\|\cdot\| \le \epsilon}(\delta_i + \eta_2 \cdot sign(\nabla_{\delta_i} G(\mathbf{w}_t, \mathbf{x}_i + \delta_i, y_i)))$, where $\Pi_\Omega(\cdot)$ is the projection operator.
9:    **end for**
10:  **end for**
11:  For each $\mathbf{x}_i \in \mathcal{B}_+$, update $\mathbf{u}_{\mathbf{x}_i}^1$ and $\mathbf{u}_{\mathbf{x}_i}^2$ by Equation (2)
12:  Update $\mathbf{u}$ by Equation (11) and compute $\widehat{\nabla}_{\mathbf{w}_t} R(\mathbf{w}_t, \delta, \mathcal{D})$ by Equation (12)
13:  Compute stochastic gradient estimator:

$$\nabla_{\mathbf{w}_t} = \widehat{\nabla}_{\mathbf{w}_t} P(\mathbf{w}_t) + \lambda(\widehat{\nabla}_{\mathbf{w}_t} R(\mathbf{w}_t, \delta, \mathcal{D}) + \frac{1}{B} \sum_{\mathbf{x}_i \in \mathcal{B}} L(h_{\mathbf{w}_t}(\mathbf{x}_i), h_{\mathbf{w}_t}(\mathbf{x}_i + \delta_i)))$$

14:  Update $\mathbf{w}_{t+1}$ by using SGD, momentum-methods or Adam.
15: **end for**

---

---
**Algorithm 4** Stochastic Algorithm for solving AdAP_PZ in (7)
---
1: Initialize $\mathbf{w}, \mathbf{u}_{\mathbf{x}}^1, \mathbf{u}_{\mathbf{x}}^2, \gamma_1$
2: **for** $t = 1, \ldots, T$ **do**
3:      Draw a batch of $B_+$ positive samples denoted by $\mathcal{B}_+$.
4:      Draw a batch of $B$ samples denoted by $\mathcal{B}$.
5:      **for** $\mathbf{x}_i \in \mathcal{B}$ **do**
6:          Initialize $\delta_i \sim \alpha \cdot \mathcal{N}(0, 1)$
7:          **for** $m = 1, \ldots, M$ **do**
8:              Update $\delta_i = \Pi_{\|\cdot\| \leq \epsilon}(\delta_i + \eta_2 \cdot sign(\nabla_{\delta_i} L(h_{\mathbf{w}_t}(\mathbf{x}_i), h_{\mathbf{w}_t}(\mathbf{x}_i + \delta_i))))$, where $\Pi_\Omega(\cdot)$ is the projection operator.
9:          **end for**
10:      **end for**
11:      For each $\mathbf{x}_i \in \mathcal{B}_+$, update $\mathbf{u}_{\mathbf{x}_i}^1$ and $\mathbf{u}_{\mathbf{x}_i}^2$ by Equation (2)
12:      Update $\mathbf{u}$ by Equation (11) and compute $\widehat{\nabla}_{\mathbf{w}_t} P(\mathbf{w}_t)$ by Equation (3)
13:      Compute stochastic gradient estimator

$$\nabla_{\mathbf{w}_t} = \widehat{\nabla}_{\mathbf{w}_t} P(\mathbf{w}_t) + \frac{\lambda}{B} \sum_{\mathbf{x}_i \in \mathcal{B}} L(h_{\mathbf{w}_t}(\mathbf{x}_i), h_{\mathbf{w}_t}(\mathbf{x}_i + \delta_i))$$

14:      Update $\mathbf{w}_{t+1}$ by using SGD, momentum-methods or Adam.
15: **end for**
---

---
**Algorithm 5** Stochastic Algorithm for solving AdAP_LZ
---
1: Initialize $\mathbf{w}, \mathbf{u}_{\mathbf{x}}^1, \mathbf{u}_{\mathbf{x}}^2, \mathbf{u}, \gamma_1, \gamma_2$
2: **for** $t = 1, \ldots, T$ **do**
3:      Draw a batch of $B_+$ positive samples denoted by $\mathcal{B}_+$.
4:      Draw a batch of $B$ samples denoted by $\mathcal{B}$.
5:      For $\mathbf{x}_i \in \mathcal{B}$, initialize $\delta_i \sim \alpha \cdot \mathcal{N}(0, 1)$
6:      **for** $m = 1, \ldots, M$ **do**
7:          Update $\mathbf{u}$ by Equation (11) and compute $\widehat{\nabla}_\delta R(\mathbf{w}_t, \delta, \mathcal{D})$ by Equation (19)
8:          Update $\delta_i = \Pi_{\|\cdot\| \leq \epsilon}(\delta_i + \eta_2 \cdot sign(\widehat{\nabla}_{\delta_i} R(\mathbf{w}_t, \delta, \mathcal{D})))$, where $\Pi_\Omega(\cdot)$ is the projection operator.
9:      **end for**
10:      For each $\mathbf{x}_i \in \mathcal{B}_+$, update $\mathbf{u}_{\mathbf{x}_i}^1$ and $\mathbf{u}_{\mathbf{x}_i}^2$ by Equation (2)
11:      Update $\mathbf{u}$ by Equation (11) and compute $\widehat{\nabla}_{\mathbf{w}_t} R(\mathbf{w}_t, \delta, \mathcal{D})$ by Equation (12)
12:      Compute stochastic gradient estimator $\nabla_{\mathbf{w}_t} = \widehat{\nabla}_{\mathbf{w}_t} P(\mathbf{w}_t) + \lambda \widehat{\nabla}_{\mathbf{w}_t} R(\mathbf{w}_t; \delta)$
13:      Update $\mathbf{w}_{t+1}$ by using SGD, momentum-methods or Adam.
14: **end for**
---

