# OpenReview forum: "Maximization of Average Precision for Deep Learning with Adversarial Ranking Robustness"
_NeurIPS.cc/2023/Conference — NeurIPS 2023 spotlight_

### Official Review · Reviewer_Szkp · 2023-07-05

**Soundness:** 3 good
**Presentation:** 3 good
**Contribution:** 2 fair
**Rating:** 5
**Confidence:** 3

**Summary:**

This paper studies the average precision issue in adversarial training. As attacking a single image may not affect the final accuracy, the average precision could be largely decreased. As a result, such a phenomenon is demonstrated to be harmful to applying adversarial training. To encourage AP robustness, a novel method is proposed by combining adversarial training and AP maximization. Additionally, by adding point-wise regularization, different variants are proposed. Through empirical analysis of many well-known datasets, the authors carefully validate the effectiveness of the proposed methods.

**Strengths:**

- This paper is well-written and can be easily understood.
- The effectiveness is great on many datasets.
- Detailed analysis of many variants of AdAP is provided.

**Weaknesses:**

- The major concern is the motivation of this paper. Why average precision is important in adversarial training is not sufficiently addressed. It seems like the research problem is ad hoc such that the proposed method could directly combine adversarial training and maximum precision. In the real world, I don’t think attacking a single example is worth investigating, and the described situation exists ubiquitously. Moreover, it is possible that in many realistic scenarios, the difference between the average accuracy and average precision might not be very large. Please justify.

- I am not sure why two regularizations AdAP_MM and AdAP_PZ should be proposed together when the first one does not have a significant advantage compared to the last one.

- The proposed method is limited to binary and imbalanced classification settings, which are strict on the problem setting. The performance of multi-class or balanced adversarial training is still questionable.

- How does the hyper-parameters $\lambda$, $\gamma_1$ and $\gamma_2$ are decided? Are they sensitive to different values?

**Questions:**

Please refer to weaknesses.

**Limitations:**

Limitations are discussed in the paper. No potential negative societal impact is found.

---

> ### Author Rebuttal · Authors · 2023-08-09
>
> Thank you for your constructive comments. Below we would like to address
> your concerns.
>
> **Q1:** About the motivation of this paper.
>
> **A:** First, we'd like to clarify that average precision is important,
> especially in scenarios with highly imbalanced datasets, e.g. medical
> diagnosis, molecular property prediction (e.g. MIT AICures challenge,
> Open Graph Benchmark) and object detection, where there could be
> thousands of negative samples but only few positive samples. Suppose we
> are required to diagnose a rare lethal disease that only 10 over 10,000
> patients suffer from. A naive model may infer all patients as negative
> to reach 99.9% accuracy. However, AP serves as a ranking metric that is
> particularly attuned to errors at the top of the ranking list, which
> makes it a more appropriate metric for reflecting models' performance
> with highly imbalanced datasets. Such a naive model achieves an AP score
> of 0, indicating it learned nothing. Second, in this paper, we are not
> limited to solving the problem that adversary is only attacking a single
> example. The reason why we use only one attacked sample in the
> introduction is to try to illustrate the importance of average precision
> with a simple example. In fact, throughout the paper, we solve the
> problems where all the input samples are attacked. In addition, we agree
> that in some realistic scenarios, the difference between the average
> accuracy and average precision might not be very large, particularly
> when the dataset is balanced. But it is important to note that such
> scenarios with balanced datasets are not the primary focus of this
> paper.
>
> **Q2:** I am not sure why two regularizations AdAP_MM and AdAP_PZ should
> be proposed together when the first one does not have a significant
> advantage compared to the last one.
>
> **A:** AdAP_MM and AdAP_PZ represent two straightforward adversarial AP
> maximization baselines by directly extending the ideas from \[Ref1\] and
> TRADES. We introduce them as baselines to contrast with our proposed
> AdAP_LN and AdAP_LPN methods since we are trying to compare our proposed
> method with other related adversarial AP maximization ideas to show the
> superiority. AdAP_LPN and AdAP_LN are proposed to show that (i) the
> proposed listwise regularization is key to improve the performance;
> (ii) neither of them is dominating the other meaning that the traditional
> pointwise regularization can help sometimes.
>
> Ref1: Madry, Aleksander, et al. Towards Deep Learning Models Resistant
> to Adversarial Attacks. ICLR 2018.
>
> **Q3:** The proposed method is limited to binary and imbalanced
> classification settings, which are strict on the problem setting. The
> performance of multi-class or balanced adversarial training is still
> questionable.
>
> **A:** There have been tons of paper proposing solutions for multi-class
> or balanced setting. However, the adversarial training method for
> imbalanced data is still under-explored. Hence, as mentioned in the
> introduction part, this paper focuses on solving imbalanced adversarial
> training problem which is meaningful and important and in which
> accuracy-based adversarial training methods are not sufficient. This
> should not be considered as a limitation but rather as a strength.
>
> **Q4:** How does the hyper-parameters $\lambda,\gamma_1, \gamma_2$, and
> are decided? Are they sensitive to different values?
>
> **A:** From Section 5.3, we can observe that $\lambda$ is a sensitive
> parameter that balances the trade-off between robustness and clean AP
> performance. This is expected. Hence, it's necessary to tune it
> carefully to achieve desired performance. The $\gamma_1, \gamma_2$ are
> tuned in $\\{0.1, 0.9\\}$ as mentioned in the paper. We have added some
> experiments on CIFAR10 and BDD100K datasets to show the sensitivity of
> these parameters below.
>
> | CIFAR10 (average) | $\gamma_2=0.1$          | $\gamma_2=0.3$          | $\gamma_2=0.5$ | $\gamma_2=0.7$ | $\gamma_2=0.9$ |
> |-------------------|-------------------------|-------------------------|----------------|----------------|----------------|
> | $\gamma_1=0.1$    | 0.2719(0.0116)          | 0.273(0.0079)           | 0.275(0.0105)  | 0.2756(0.0135) | 0.2753(0.0114) |
> | $\gamma_1=0.3$    | 0.2701(0.012)           | 0.2696(0.01)            | 0.2753(0.0128) | 0.2733(0.0101) | 0.2745(0.0107) |
> | $\gamma_1=0.5$    | 0.2674(0.0088)          | 0.2741(0.011)           | 0.2688(0.0108) | 0.2685(0.0135) | 0.2741(0.0113) |
> | $\gamma_1=0.7$    | 0.2649(0.0103)          | 0.2713(0.007)           | 0.2686(0.0165) | 0.2696(0.0138) | 0.2658(0.0103) |
> | $\gamma_1=0.9$    | 0.2668(0.0137)          | **0.2766(0.0131)**   | 0.2668(0.013)  | 0.2689(0.0139) | 0.2641(0.0124) |
> | **BDD100K(rainy)**  | $\gamma_2=0.1$          | $\gamma_2=0.3$          | $\gamma_2=0.5$ | $\gamma_2=0.7$ | $\gamma_2=0.9$ |
> | $\gamma_1=0.1$    | 0.2433(0.0209)          | 0.2436(0.0205)          | 0.2404(0.0188) | 0.2415(0.0220) | 0.2423(0.0215) |
> | $\gamma_1=0.3$    | 0.2471(0.0219)          | 0.2440(0.0174)          | 0.2432(0.0169) | 0.2473(0.0214) | 0.2460(0.0188) |
> | $\gamma_1=0.5$    | 0.2479(0.0210)          | 0.2475(0.0209)          | 0.2476(0.0217) | 0.2450(0.0189) | 0.2472(0.0187) |
> | $\gamma_1=0.7$    | 0.2507(0.0203)          | 0.2453(0.0204)          | 0.2471(0.0179) | 0.2461(0.0206) | 0.2480(0.0176) |
> | $\gamma_1=0.9$    | **0.2522(0.0208)**    | 0.2479(0.0200)          | 0.2485(0.0201) | 0.2447(0.0193) | 0.2478(0.0193) |

---

> > ### Comment · Reviewer_Szkp · 2023-08-19
> > **Reply**
> >
> > Thanks for the rebuttal, I have checked your answer which addresses most of my concerns. I decided to raise my score to 5.

---

### Official Review · Reviewer_YhER · 2023-07-05

**Soundness:** 4 excellent
**Presentation:** 4 excellent
**Contribution:** 3 good
**Rating:** 8
**Confidence:** 5

**Summary:**

The paper focuses on adversarial training in terms of Average Precision (AP), which is guided by three design principles: trade-off between AP and robustness, robustness in terms of AP instead of accuracy, and consistency of attacks. By utilizing the techniques of stochastic compositional optimization, the paper proposes a series of adversarial training algorithms to handle the inter-dependent perturbations.

**Strengths:**

1.	Novelty: To the best of our knowledge, it is the first work to consider adversarial training of AP. It is a non-trivial extension due to the non-decomposable formulation of AP.

2.	Significance: As a widely-used ranking metric, the robustness of AP is significant to the machine learning community. Besides, the design principles and techniques might be instructive to the robustness of other ranking metrics.

3.	Clarity: The paper is overall well-written with clear notations.

4.	Soundness: The effectiveness of the proposed method is well-supported by experiments under various settings.


**Weaknesses:**

1.	The authors solve a non-zero-sum game to ensure consistency. However, unlike previous work on adversarial training, the equilibrium state of this game is unknown and requires more discussion.

2.	Fig. 2 provides a visualization of the trade-off between robustness and AP, but how the hyperparameter $\lambda$ affects the trade-off is unclear. Ideally, it should present a positive correlation.

3.	The related work could be further improved by discussing the latest literature on AP stochastic optimization such as [1,2].

Ref:

[1] Wang et. al. Momentum accelerates the convergence of stochastic auprc maximization. ICML, 2022.

[2] Wen et. al. Exploring the algorithm-dependent generalization of auprc optimization with list stability. NeurIPS, 2022.


**Questions:**

Please refer to the weaknesses part for the major concerns. Other minor issues are as follows:

1.	The design of $R$ in Eq. (8) requires a detailed explanation: AP involves all examples, while the top-one probability focuses on the top-one examples. Could we apply ranking-based functions instead?

2.	The proposed algorithms share similar properties with [19] and [32]. Is it possible to provide a corresponding convergence analysis based on these works?


**Limitations:**

Limitations are addressed.

---

> ### Author Rebuttal · Authors · 2023-08-09
>
> We first address what the reviewer mentioned as weaknesses and then
> respond to the reviewer's questions:
>
> **Q1:** The authors solve a non-zero-sum game to ensure consistency.
> However, unlike previous work on adversarial training, the equilibrium
> state of this game is unknown and requires more discussion.
>
> **A:** We appreciate your comments. We acknowledge that this is a
> separate theoretical issue. Even for zero-sum game approaches, the
> non-convexity nature of the problem may render the Nash equilibrium
> non-existent \[Ref1\]. We hope our research will motivate more
> theoretical researchers to study this problem.
>
> Ref1: Jin et al. What is Local Optimality in Nonconvex-Nonconcave
> Minimax Optimization. ICML, 2020.
>
> **Q2:** Fig. 2 provides a visualization of the trade-off between
> robustness and AP, but how the hyperparameter affects the trade-off is
> unclear. Ideally, it should present a positive correlation.
>
> **A:** As the curves shown in Fig.2, it does presents a positive
> correlation between robust AP and $\lambda$ for relatively small values
> of $\lambda$. Nevertheless, as $\lambda$ becomes excessively large, this
> correlation diminishes. This reasoning is logical to us since we might
> expect a meaningless model if $\lambda$ goes to infinity.
>
> **Q3:** The related work could be further improved by discussing the
> latest literature on AP stochastic optimization such as \[1,2\].
>
> **A:** Thank you for your suggestion. we will incorporate discussion of
> the suggested literature to enhance the related work section in the
> revised paper.
>
> **Q4:** The design of in Eq. (8) requires a detailed explanation: AP
> involves all examples, while the top-one probability focuses on the
> top-one examples. Could we apply ranking-based functions instead?
>
> **A:** Indeed, the regularization based on the top-one probability
> $p(x_i) = \frac{\exp(h_w(x_i))}{\sum_{j=1}^n\exp(h_w(x_j))}$
> is a ranking-based function. Please note that the regularization sums
> over all data instead of just one example. In addition, the denominator
> of the top-one probability also includes the all example, similar to AP.
> A similar loss was originally proposed in \[Ref2\] known as ListNet for
> learning to rank. It encourages elevating positive samples to higher
> positions in the list as opposed to negative
>
> Ref2: Cao et al. Learning to rank: from pairwise approach to listwise
> approach. ICML 2007.
>
> **Q5:** The proposed algorithms share similar properties with \[19\] and
> \[32\]. Is it possible to provide a corresponding convergence analysis
> based on these works?
>
> **A:** Please note that our problem is much more challenging than that
> in \[19, 32\]. We can also view the problem as a bilevel optimization
> problem. However, the lower-level problem itself is non-convex. Almost
> all existing convergence analysis for bilevel optimization assume
> convexity for the lower-level problem. Deriving the convergence of our
> algorithm and any other algorithms would be a significant work by
> itself.

---

> > ### Comment · Reviewer_YhER · 2023-08-13
> >
> > Thank you for your responses. While some concerns have been addressed, the following issues are still unclear:
> >
> > **Q2**: Fig. 2 only presents a positive correlation between **robust AP and clean AP** instead of **robust AP and $\lambda$**. Please plot "robust AP v.s. $\lambda$" and "clean AP v.s. $\lambda$" respectively to support the conclusions.
> >
> > **Q4**: Although both the top-one probability and AP involve all examples, it will be better if more theoretical derivations are provided. In fact, most ranking losses involve all examples such as NDCG. Compared with other ranking functions, the advantages of the top-one probability are unclear.
> >
> > **Q5**: The theoretical convergence analysis is indeed challenging. However, it is necessary to provide an empirical analysis since the paper proposes a new optimization algorithm.

---

> > > ### Author Response · Authors · 2023-08-15
> > >
> > > Thank you for the prompt feedback and we hope to address the remaining
> > > issues.
> > >
> > > **Q2:** Fig. 2 only presents a positive correlation between robust AP
> > > and clean AP instead of robust AP and $\lambda$ . Please plot \"robust
> > > AP v.s. $\lambda$ \" and \"clean AP v.s. $\lambda$ \" respectively to
> > > support the conclusions.
> > >
> > > **A:** For illustration purposes, we provide the relationship between
> > > robust AP/clean AP and $\lambda$ on CIFAR10_cls1, CIFAR100_cls2,
> > > BDD100K(cloudy), CelebA(gray_hair), corresponding to Fig.2 shown in the
> > > paper. The results are summarized in the table below since the system
> > > doesn't support image uploading. We report the results in the format of robust AP/clean AP, with the values representing the average of three runs with different seeds. From the table, we can observe that
> > > there is a positive correlation between robust AP and $\lambda$ for
> > > relatively small values of $\lambda$ and this correlation diminishes
> > > when $\lambda$ continues increasing. Moreover, a negative correlation
> > > between clean AP and $\lambda$ is consistently observed, in accordance
> > > with our expectations. We will include the results in the revision.
> > >
> > > |$ \lambda  $          | 0.01          | 0.4           | 0.8           | 1             | 4             | 8             | 10            |
> > > |------------------|---------------|---------------|---------------|---------------|---------------|---------------|---------------|
> > > | CIFAR10 | (cls\_1)
> > > | AdAP\_LN           | 0.1153/0.9525 | 0.4264/0.9295 | 0.4590/0.9190 | 0.4464/0.9099 | 0.3181/0.8649 | 0.2695/0.8385 | 0.2522/0.8356 |
> > > | AdAP\_LPN          | 0.1504/0.9358 | 0.5185/0.9314 | 0.5263/0.9146 | 0.5229/0.9067 | 0.4818/0.8468 | 0.4517/0.8102 | 0.4411/0.7968 |
> > > | CIFAR100  | (cls\_2)
> > > | AdAP\_LN           | 0.0811/0.8752 | 0.2782/0.7984 | 0.3184/0.7668 | 0.3238/0.7540 | 0.3644/0.6736 | 0.3437/0.6286 | 0.3436/0.6164 |
> > > | AdAP\_LPN          | 0.1604/0.8332 | 0.3475/0.7896 | 0.3809/0.7558 | 0.3859/0.7441 | 0.3724/0.6470 | 0.3501/0.6011 | 0.3392/0.5827 |
> > > | BDD100K   | (cloudy)
> > > | AdAP\_LN           | 0.0621/0.6504 | 0.2221/0.5986 | 0.2742/0.5632 | 0.2815/0.5577 | 0.2796/0.4935 | 0.2347/0.4937 | 0.2103/0.4861 |
> > > | AdAP\_LPN          | 0.0411/0.6570 | 0.2455/0.5985 | 0.2682/0.5681 | 0.2726/0.5600 | 0.2725/0.4886 | 0.2464/0.4366 | 0.2473/0.4295 |
> > > | CelebA| (gray\_hair)
> > > | AdAP\_LN           | 0.0264/0.7037 | 0.1831/0.6282 | 0.2449/0.6061 | 0.2696/0.6010 | 0.2859/0.5221 | 0.2714/0.4662 | 0.2601/0.4506 |
> > > | AdAP\_LPN          | 0.0284/0.7123 | 0.2280/0.6259 | 0.2555/0.5907 | 0.2644/0.5804 | 0.2799/0.4948 | 0.2676/0.4515 | 0.2676/0.4515 |
> > >
> > >
> > >
> > > **Q4:** Although both the top-one probability and AP involve all
> > > examples, it will be better if more theoretical derivations are
> > > provided. In fact, most ranking losses involve all examples such as
> > > NDCG. Compared with other ranking functions, the advantages of the
> > > top-one probability are unclear.
> > >
> > > **A:** (1) NDCG is not appropriate for defining the regularization term.
> > > Please note that our regularization needs to measure the divergence
> > > between **real-valued ranking scores** of two sets of data, i.e., that
> > > of the clean data and the perturbed data. NDCG measures the consistency
> > > between **the real-valued ranking scores and ground-truth discrete
> > > relevance scores**, making it not appropriate for our purpose. (2) The
> > > top-one probabilities springing from learning to ranking literature
> > > provides a natural way for measuring the divergence between two
> > > probability distributions, making it able to characterize the boundary
> > > error as derived in Theorem 1. (3) Other ranking correlation measures
> > > (e.g., Spearman's $\rho$, Kendall's $\tau$) are possible to measure the
> > > difference between two ranking results. Nevertheless, the advantage of
> > > using top-one probabilities is that it does not involve all pairs of
> > > data and its optimization is much more efficient.
> > >
> > > **Q5:** The theoretical convergence analysis is indeed challenging.
> > > However, it is necessary to provide an empirical analysis since the
> > > paper proposes a new optimization algorithm.
> > >
> > > **A:** Thank you for your valuable suggestions. We will provide some
> > > empirical results showing the convergence of our algorithms in the
> > > revision. Below we show some convergence results of our proposed AdAP_LN
> > > algorithm. Specifically, we set $\lambda=1, \gamma_1=0.1, \gamma_2=0.1$
> > > and run AdAP_LN algorithm on CIFAR10 dataset and BDD100K dataset for a
> > > total of 120 epochs and 80 epochs, respectively. We evaluate the
> > > training loss after each epoch and report the loss values below, as
> > > image uploading is not supported by the system. We present the AP loss
> > > (i.e., $P(w)$ in Equation 9) and regularization term ( i.e.,
> > > $R(w,\delta,D)$ in Equation 9) separately, as well as the summation of
> > > the two losses. For each experiment, we repeat three times with
> > > different random seeds, then report average loss values. The results
> > > demonstrate the convergence of our algorithm. We will include plots of
> > > convergence curves in the revision.

---

> > > > ### Author Response · Authors · 2023-08-15
> > > > **Appended Results_Part1**
> > > >
> > > > -   CIFAR10_cls0
> > > >
> > > >     -   $P(w)$: \[-0.2964 -0.3067 -0.3123 -0.3167 -0.3279 -0.3348
> > > >         -0.3326 -0.3369 -0.3475 -0.3593 -0.3491 -0.3527 -0.3451 -0.3615
> > > >         -0.3685 -0.3569 -0.3728 -0.3557 -0.3748 -0.3653 -0.3624 -0.3721
> > > >         -0.3719 -0.3735 -0.3892 -0.3762 -0.3861 -0.382 -0.3643 -0.3942
> > > >         -0.3902 -0.3809 -0.3968 -0.3908 -0.3868 -0.3861 -0.3905 -0.405
> > > >         -0.4099 -0.3801 -0.388 -0.3939 -0.3974 -0.3953 -0.4071 -0.4156
> > > >         -0.4017 -0.4038 -0.4158 -0.4008 -0.406 -0.4084 -0.4104 -0.4023
> > > >         -0.4062 -0.4165 -0.4172 -0.4296 -0.4037 -0.4161 -0.4401 -0.453
> > > >         -0.4585 -0.4564 -0.4671 -0.4552 -0.4597 -0.4631 -0.4632 -0.4677
> > > >         -0.4657 -0.4685 -0.4703 -0.4739 -0.4736 -0.4792 -0.4815 -0.4793
> > > >         -0.4764 -0.484 -0.4715 -0.4793 -0.4726 -0.478 -0.485 -0.4764
> > > >         -0.4842 -0.4888 -0.4829 -0.4906 -0.489 -0.4954 -0.4976 -0.496
> > > >         -0.4916 -0.4938 -0.4955 -0.4955 -0.4966 -0.4946 -0.4956 -0.4999
> > > >         -0.4937 -0.4969 -0.4954 -0.5 -0.4956 -0.4966 -0.4997 -0.496
> > > >         -0.4998 -0.4956 -0.4995 -0.4957 -0.4925 -0.4994 -0.502 -0.4991
> > > >         -0.5034 -0.5057\]
> > > >
> > > >     -   $R(w,\delta,D)$: \[0.0745 0.055 0.0484 0.0436 0.0457 0.0435
> > > >         0.041 0.0397 0.0428 0.0486 0.0454 0.0445 0.0424 0.0432 0.0459
> > > >         0.0438 0.0496 0.04 0.0494 0.0428 0.0455 0.0498 0.053 0.0473
> > > >         0.0502 0.0462 0.0457 0.0488 0.0466 0.0523 0.0478 0.0457 0.0496
> > > >         0.0521 0.046 0.0449 0.0463 0.0522 0.0548 0.0446 0.0493 0.0493
> > > >         0.0488 0.0491 0.0571 0.0534 0.0522 0.0491 0.0526 0.0497 0.0507
> > > >         0.0494 0.052 0.0488 0.054 0.054 0.0567 0.0575 0.0463 0.0539
> > > >         0.0542 0.0561 0.0573 0.0553 0.0569 0.0538 0.0555 0.0563 0.058
> > > >         0.0586 0.06 0.0591 0.0606 0.0579 0.0584 0.0596 0.0603 0.0605
> > > >         0.0598 0.0603 0.0577 0.0606 0.0581 0.0594 0.0612 0.0621 0.0634
> > > >         0.0629 0.0621 0.0655 0.0642 0.0653 0.0667 0.0636 0.0608 0.0652
> > > >         0.0646 0.0633 0.0652 0.0647 0.0649 0.0656 0.0652 0.0656 0.0644
> > > >         0.0651 0.0655 0.0645 0.0654 0.0659 0.0656 0.0653 0.0662 0.065
> > > >         0.0641 0.0668 0.0677 0.0662 0.0663 0.0677\]
> > > >
> > > >     -   $P(w)+R(w,\delta,D)$: \[-0.2219 -0.2517 -0.2639 -0.2731
> > > >         -0.2821 -0.2913 -0.2917 -0.2972 -0.3047 -0.3107 -0.3037 -0.3082
> > > >         -0.3028 -0.3183 -0.3225 -0.3132 -0.3232 -0.3157 -0.3255 -0.3225
> > > >         -0.3168 -0.3223 -0.3189 -0.3262 -0.339 -0.3299 -0.3404 -0.3332
> > > >         -0.3177 -0.3419 -0.3424 -0.3352 -0.3472 -0.3388 -0.3408 -0.3412
> > > >         -0.3442 -0.3528 -0.3551 -0.3355 -0.3386 -0.3446 -0.3486 -0.3461
> > > >         -0.35 -0.3622 -0.3495 -0.3547 -0.3632 -0.3511 -0.3552 -0.359
> > > >         -0.3584 -0.3534 -0.3522 -0.3625 -0.3605 -0.3721 -0.3574 -0.3622
> > > >         -0.3859 -0.3969 -0.4012 -0.4011 -0.4102 -0.4014 -0.4042 -0.4068
> > > >         -0.4052 -0.4091 -0.4057 -0.4094 -0.4098 -0.416 -0.4152 -0.4196
> > > >         -0.4213 -0.4188 -0.4167 -0.4236 -0.4138 -0.4187 -0.4145 -0.4185
> > > >         -0.4239 -0.4143 -0.4208 -0.4259 -0.4208 -0.4251 -0.4247 -0.4302
> > > >         -0.431 -0.4324 -0.4308 -0.4286 -0.4308 -0.4321 -0.4314 -0.4299
> > > >         -0.4307 -0.4343 -0.4285 -0.4313 -0.431 -0.4348 -0.4302 -0.432
> > > >         -0.4344 -0.4301 -0.4342 -0.4303 -0.4333 -0.4307 -0.4284 -0.4326
> > > >         -0.4343 -0.4329 -0.4371 -0.438 \]

---

> > > > > ### Author Response · Authors · 2023-08-15
> > > > > **Appended Results_Part2**
> > > > >
> > > > > -   CIFAR10_cls1
> > > > >
> > > > >     -   $P(w)$: \[-0.3372 -0.3665 -0.361 -0.3801 -0.3922 -0.3824 -0.3956 -0.3908 -0.4065 -0.4263 -0.4143 -0.4347 -0.4365 -0.46 -0.4449 -0.4651 -0.4525 -0.4513 -0.4715 -0.4824 -0.4713 -0.4767 -0.4625 -0.4839 -0.4996 -0.5041 -0.4862 -0.4702 -0.4892 -0.5158 -0.492 -0.488 -0.4714 -0.528 -0.517 -0.5068 -0.5166 -0.5039 -0.5044 -0.5201 -0.5214 -0.5217 -0.5249 -0.5202 -0.5213 -0.5453 -0.5223 -0.5163 -0.5488 -0.5206 -0.5469 -0.5501 -0.5379 -0.5582 -0.5256 -0.5165 -0.5534 -0.556 -0.525 -0.5224 -0.5567 -0.5655 -0.5752 -0.5823 -0.5739 -0.5835 -0.5892 -0.5946 -0.6072 -0.5945 -0.5997 -0.6122 -0.5878 -0.6071 -0.6092 -0.6136 -0.6048 -0.6212 -0.6143 -0.6115 -0.6106 -0.6106 -0.6126 -0.6187 -0.6264 -0.613 -0.6189 -0.639 -0.6282 -0.6365 -0.6329 -0.6369 -0.6432 -0.6388 -0.6349 -0.6328 -0.6305 -0.6445 -0.6535 -0.6479 -0.6524 -0.6442 -0.6458 -0.6491 -0.6499 -0.6489 -0.6493 -0.6453 -0.6466 -0.6455 -0.6454 -0.651 -0.6427 -0.6469 -0.6464 -0.6477 -0.6529 -0.6433 -0.6464 -0.6508\]
> > > > >
> > > > >     -   $R(w,\delta,D)$: \[0.1483 0.1273 0.105 0.1081 0.1022 0.09 0.0968 0.0914 0.1051 0.1035 0.0995 0.1061 0.1008 0.1111 0.1006 0.1012 0.1005 0.1083 0.1028 0.1092 0.1049 0.1052 0.1022 0.1042 0.1089 0.1045 0.1052 0.1018 0.0981 0.1143 0.1034 0.1045 0.116 0.1114 0.117 0.1097 0.1094 0.1037 0.1037 0.1165 0.0991 0.108 0.1093 0.1101 0.1053 0.1163 0.1186 0.1089 0.1241 0.1102 0.1194 0.1195 0.1103 0.1199 0.1058 0.1089 0.1174 0.1229 0.1105 0.1106 0.1037 0.102 0.104 0.1045 0.1011 0.1016 0.1079 0.1038 0.1079 0.1025 0.1054 0.1054 0.1017 0.1032 0.1077 0.1095 0.0883 0.1054 0.1028 0.1088 0.1057 0.1056 0.0946 0.1004 0.106 0.1104 0.1081 0.1092 0.1081 0.1089 0.1074 0.1053 0.1076 0.1079 0.1047 0.1053 0.1041 0.1027 0.1091 0.1038 0.1075 0.108 0.1093 0.1087 0.1041 0.1092 0.1038 0.1054 0.1046 0.1087 0.1074 0.1115 0.1075 0.1092 0.1091 0.1098 0.1089 0.1081 0.1072 0.1084\]
> > > > >
> > > > >     -   $P(w)+R(w,\delta,D)$: \[-0.1889 -0.2391 -0.256 -0.272 -0.2899 -0.2923 -0.2988 -0.2994 -0.3014 -0.3228 -0.3148 -0.3286 -0.3357 -0.3489 -0.3442 -0.3639 -0.352 -0.343 -0.3686 -0.3732 -0.3665 -0.3715 -0.3603 -0.3797 -0.3907 -0.3997 -0.381 -0.3684 -0.3911 -0.4016 -0.3886 -0.3834 -0.3554 -0.4166 -0.4 -0.3971 -0.4072 -0.4002 -0.4008 -0.4037 -0.4222 -0.4137 -0.4157 -0.41 -0.416 -0.4291 -0.4037 -0.4075 -0.4247 -0.4105 -0.4275 -0.4306 -0.4276 -0.4383 -0.4197 -0.4077 -0.4361 -0.433 -0.4145 -0.4118 -0.4531 -0.4634 -0.4712 -0.4778 -0.4728 -0.4819 -0.4814 -0.4908 -0.4993 -0.492 -0.4942 -0.5068 -0.4862 -0.5039 -0.5015 -0.5042 -0.5165 -0.5158 -0.5114 -0.5027 -0.5049 -0.5049 -0.518 -0.5183 -0.5204 -0.5026 -0.5108 -0.5298 -0.52 -0.5277 -0.5255 -0.5316 -0.5356 -0.5309 -0.5303 -0.5274 -0.5263 -0.5418 -0.5445 -0.5442 -0.5448 -0.5362 -0.5365 -0.5404 -0.5458 -0.5398 -0.5456 -0.5399 -0.542 -0.5369 -0.538 -0.5395 -0.5353 -0.5377 -0.5372 -0.5379 -0.544 -0.5352 -0.5392 -0.5424\]
> > > > >
> > > > > -   BDD100K(rainy)
> > > > >
> > > > >     -   $P(w)$: \[-0.1606 -0.1644 -0.1654 -0.1657 -0.1749 -0.1774
> > > > >         -0.1792 -0.1813 -0.1871 -0.1921 -0.2034 -0.1978 -0.1953 -0.2074
> > > > >         -0.1986 -0.2103 -0.2108 -0.2125 -0.2075 -0.2193 -0.2214 -0.2268
> > > > >         -0.2272 -0.2262 -0.2311 -0.2204 -0.2305 -0.2357 -0.2317 -0.2355
> > > > >         -0.24 -0.2401 -0.2314 -0.2464 -0.2455 -0.2478 -0.2481 -0.2489
> > > > >         -0.2525 -0.2563 -0.2713 -0.2804 -0.2744 -0.2763 -0.2867 -0.2856
> > > > >         -0.2914 -0.2884 -0.2921 -0.2907 -0.2981 -0.2992 -0.2971 -0.3005
> > > > >         -0.3006 -0.3041 -0.3031 -0.3051 -0.3118 -0.3045 -0.3119 -0.3128
> > > > >         -0.3132 -0.3144 -0.3125 -0.3161 -0.3156 -0.316 -0.3159 -0.3193
> > > > >         -0.318 -0.3162 -0.3159 -0.3213 -0.3228 -0.3171 -0.3187 -0.3192
> > > > >         -0.3242 -0.3201\]
> > > > >
> > > > >     -   $R(w,\delta,D)$: \[0.0686 0.0467 0.0458 0.0489 0.0427 0.0421
> > > > >         0.0372 0.0409 0.0438 0.0449 0.0485 0.0465 0.0391 0.0416 0.0379
> > > > >         0.0442 0.0428 0.0471 0.0418 0.0468 0.0499 0.0501 0.0469 0.0469
> > > > >         0.0467 0.0438 0.0468 0.0467 0.0467 0.0513 0.0513 0.0463 0.0446
> > > > >         0.0496 0.0504 0.0521 0.0505 0.0526 0.0466 0.053 0.0488 0.0507
> > > > >         0.0502 0.0518 0.0518 0.0514 0.0528 0.0535 0.0537 0.0531 0.0549
> > > > >         0.0551 0.0571 0.0567 0.0568 0.0574 0.0558 0.058 0.0586 0.0593
> > > > >         0.0595 0.0574 0.0581 0.0585 0.0584 0.0583 0.0576 0.0584 0.0587
> > > > >         0.0583 0.0589 0.0587 0.058 0.0597 0.0595 0.0593 0.058 0.0573
> > > > >         0.0588 0.0594\]
> > > > >
> > > > >     -   $P(w)+R(w,\delta,D)$: \[-0.092 -0.1177 -0.1197 -0.1168
> > > > >         -0.1322 -0.1353 -0.142 -0.1404 -0.1433 -0.1471 -0.1549 -0.1513
> > > > >         -0.1562 -0.1658 -0.1607 -0.1661 -0.168 -0.1654 -0.1657 -0.1725
> > > > >         -0.1714 -0.1767 -0.1803 -0.1793 -0.1843 -0.1767 -0.1837 -0.189
> > > > >         -0.185 -0.1842 -0.1888 -0.1938 -0.1868 -0.1968 -0.1951 -0.1957
> > > > >         -0.1976 -0.1962 -0.2059 -0.2032 -0.2225 -0.2297 -0.2242 -0.2245
> > > > >         -0.2349 -0.2341 -0.2386 -0.2349 -0.2384 -0.2376 -0.2432 -0.2441
> > > > >         -0.24 -0.2438 -0.2438 -0.2467 -0.2473 -0.2471 -0.2533 -0.2452
> > > > >         -0.2524 -0.2554 -0.2551 -0.2559 -0.2541 -0.2578 -0.258 -0.2576
> > > > >         -0.2572 -0.261 -0.259 -0.2575 -0.2579 -0.2616 -0.2633 -0.2578
> > > > >         -0.2607 -0.2619 -0.2654 -0.2607\]

---

### Official Review · Reviewer_N7Ak · 2023-07-06

**Soundness:** 3 good
**Presentation:** 3 good
**Contribution:** 3 good
**Rating:** 6
**Confidence:** 4

**Summary:**

This paper considers the adversarial robustness of the AP metric, which is an important measure of deep learning under some imbalanced applications. To do this, the authors develop a novel formulation that combines an AP surrogate loss with a regularization term toward adversarial ranking robustness, maintaining the consistency between the ranking of clean data and that of perturbed data. Empirical studies demonstrate the effectiveness of the proposed methods.

**Strengths:**

To the best of our knowledge, this is the first work to consider the AP-based adversarial robustness problem, which will bring some new insights to the adversarial robustness community.
The contributions of this paper are novel and the theoretical results are technically sound.
The empirical results are also promising.

**Weaknesses:**

However, some essential issues should be fixed:
1. During the evaluation, this paper merely considers the simple FGSM-based attack manner, which is insufficient to support the effectiveness of the proposed method. Some stronger attacks, such as PGD-based and AutoAttack [1], should be considered.
2. Another minor question is how AP-based AT impacts the performance of accuracy-based AT. Can the proposed methods improve AdAP without sacrificing overall accuracy? Because merely considering AP while overlooking accuracy may be meaningless.
3. Why do we need to develop AdAP? What are the differences between AdAP and AdAUC in the ranking performance? Please give me some intuitive examples like Fig.1
4. Finally, some latest advanced AP optimization methods are missed, such as [2].


Ref:
[1] Reliable Evaluation of Adversarial Robustness with an Ensemble of Diverse Parameter-free Attacks.
[2] Exploring the algorithm-dependent generalization of AUPRC optimization with list stability.

**Questions:**

Please carefully address all my concerns in the weakness part.

**Limitations:**

The authors do not include any limitations for their work.

---

> ### Author Rebuttal · Authors · 2023-08-09
>
> We thank the reviewer for dedicating their time to provide a
> comprehensive review, and we are committed to addressing the raised
> issues.
>
> **Q1:** Adding some stronger attacks, such as PGD-based and AutoAttack.
>
> **A:** We appreciate the reviewer's suggestion. First, we'd like to
> apologize for the confusion caused by the terminology 'FGSM attack'. In
> all our experiments, we utilized the iterative FGSM attack method which
> indeed works as a $l_{\infty}$-bounded PGD attack. Following \[Ref2\],
> we abused the terminology in Section 5. To provide a more comprehensive
> assessment, we have evaluated all the models against a strong attack
> method $Auto-PGD_{CE}$ proposed in \[Ref1\]. The results are summarized in Table 1 in the global response(PDF file). We can observe that $Auto-PGD_{CE}$ exhibits a stronger attack as
> it leads to the deterioration of performance across all models, compared
> with the iterative FGSM (standard PGD) method employed in section 5.1.
> However, we can see that the superiority of our proposed methods still
> remains evident. Please note that we did not compare with the ensemble
> AutoAttack since it is tailored for multi-class classification
> concerning accuracy. For instance, the $APGD_{DLR}$ approach presented
> in \[Ref1\] requires at least three classes. We focus on AP maximization
> on imbalanced binary classification data and the multi-class datasets
> such as CIFAR-10 are converted into multiple one-vs-all binary
> classification tasks with reported performance averaged over these
> multiple tasks.
>
> Ref1: Croce & Hein. Reliable evaluation of adversarial robustness with
> an ensemble of diverse parameter-free attacks. ICML 2020.
>
> Ref2: Zhang et al. Theoretically principled trade-off between robustness
> and accuracy. International conference on machine learning. ICML 2019.
>
> **Q2:** Another minor question is how AP-based AT impacts the
> performance of accuracy-based AT. Can the proposed methods improve AdAP
> without sacrificing overall accuracy?
>
> **A:** Please note that in many applications (e.g., learning to rank,
> imbalanced classification), AP is much more informative than accuracy.
> For example, in imbalanced classification with 99% data being negative,
> achieving 99% accuracy by a naive model that predicts everything as
> negative is meaningless. Hence, the focus of the paper is to report AP.
> Nevertheless, we have evaluated the performance of accuracy of our
> models on CIFAR10 dataset. To this end, we need to learn a threshold on
> the validation data since our model is a ranking-based model. Based on
> our model and the threshold, we have also evaluated our models in terms
> of accuracy and compared them with other methods in Table 2 in the global response(PDF file). The
> results show that (i) our proposed methods improve the AdAP and
> adversarial accuracy at the same time; (ii) all adversarial training
> methods' accuracy cluster around 0.9, the ratio of negative samples,
> offering limited insight into the model's performance.
>
> **Q3:** Why do we need to develop AdAP? What are the differences between
> AdAP and AdAUC in the ranking performance? Please give me some intuitive
> examples like Fig.1
>
> **A:** In scenarios involving highly imbalanced datasets, AP, as
> demonstrated in \[Ref1\], offers a more accurate reflection of the
> model's performance as compared to AUC. For example, we have a dataset
> that contains 10 positive samples and 10 thousand negative samples(e.g.
> document retrieval or object detection scenarios). Suppose one model
> ranks 10 positives higher than all the negatives, then both AP and AUC
> achieve 1. After perturbation, if the model ranks 2 negatives at the top
> and keeps the rest lower than all the positives, the AUC score will
> remain at 0.9998 while the AP score will degrade to 0.8333. This
> illustrates that, in such case, AP metric could be more informative
> about performance.
>
> Ref1: Davis and Goadrich. The relationship between Precision-Recall and
> ROC curves. ICML 2006.
>
> **Q4:** Adding citations to advanced AP optimization methods, such as
> \[2\].
>
> **A:** Thank you for your suggestion for improving our paper. we will
> include and discuss the suggested advanced AP optimization methods in
> the revision.

---

> > ### Comment · Reviewer_N7Ak · 2023-08-20
> > **Thanks for your rebuttal**
> >
> > Thank you for your effort. I think most of my concerns have been addressed.

---

### Official Review · Reviewer_rUtg · 2023-07-07

**Soundness:** 3 good
**Presentation:** 3 good
**Contribution:** 3 good
**Rating:** 6
**Confidence:** 4

**Summary:**

This paper extends the discussion of adversarial robustness from accuracy to precision, and also extends TRADES solution to this new setting. The paper is fairly standard, with a new problem, a new solution, some minor theoretical studies (obviously also extended from TRADES), and some fairly good empirical results. The paper is highly condensed, so several critical points need further clarification.

**Strengths:**

- The paper interestingly studies a new problem of precision in the adversarial setting.

- The paper introduces a new algorithm to achieve the listwise regularizations

**Weaknesses:**

1. the empirical method and theoretical discussions are extended from TRADES, which might raise some concerns about the novel contributions of this work.

2. the experiments are only conducted against FGSM attack, this is probably too limited, especially since there are several cases the performances are fairly close.
    - please also use PGD and autoAttack.

3. the an essential step of the algorithm is the approximation in Section 4.2, it's probably necessary to offer more empirical results on this regard, such as ablation studies with varying batch sizes.

**Questions:**

1. it seems the authors forgot to define AdAP_LPN properly.
    - the only clue I can find is at line 337, where AdAP_LPN is combining listwise and pointwise adversarial regularization, then in contrast AdAP_LN is probably just listwise regularization.

2. cannot find information about the threat model behind TRADES, and PGD, cannot confirm whether these threat models (models used to generated adversarial attacks) are the same as the AdAP family models or not.

**Limitations:**

do not find explicit discussions.

---

> ### Author Rebuttal · Authors · 2023-08-09
>
> Thank you for your comments and feedback on our paper. In the following,
> we are committed to addressing the raised concerns and questions.
>
> **Q1:** About the novel contributions of this work.
>
> **A:** We agree the adversarial regularization is similarly motivated as
> TRADES, which has been widely used for adversarial training. However,
> there are some key differences between our work and TRADES in how the
> regularization and attack are set up. (1) the regularization in TRADES
> measures the pointwise difference between predictions on clean data and
> perturbed data; instead the regularization in our method measures the
> listwise difference between predictions on clean data and perturbed
> data; (2) the attacks in the training of TRADES are generated in a
> zero-sum framework to maximize the regularization term; in contrast, the
> attacks in our training are generated in a non-zero sum framework. These
> two differences are very important for improving the performance of
> adversarial AP. In addition, the optimization of the proposed listwise
> regularization, maintaining the consistency between the ranking of clean
> data and that of perturbed data, is a non-trivial extension of TRADES.
>
> **Q2:** Adding more attacks, e.g., PGD and AutoAttack.
>
> **A:** We appreciate the reviewer's suggestion. First, we'd like to
> apologize for the confusion caused by the terminology 'FGSM attack'. In
> all our experiments, we utilized the iterative FGSM attack method which
> indeed works as a $l_{\infty}$-bounded PGD attack. Following \[Ref2\],
> we abused the terminology in Section 5. To provide a more comprehensive
> assessment, we have evaluated all the models against a strong attack
> method $Auto-PGD_{CE}$ proposed in \[Ref1\]. The results
> are summarized in Table 1 in the global response(PDF file). We can observe that $Auto-PGD_{CE}$ exhibits a
> stronger attack as it leads to the deterioration of performance across
> all models, compared with the iterative FGSM (standard PGD) method
> employed in section 5.1. However, we can see that the superiority of our
> proposed methods still remains evident. Please note that we did not
> compare with the ensemble AutoAttack since it is tailored for
> multi-class classification concerning accuracy. For instance, the
> $APGD_{DLR}$ approach presented in \[Ref1\] requires at least
> three classes. We focus on AP maximization on imbalanced binary
> classification data and the multi-class datasets such as CIFAR-10 are
> converted into multiple one-vs-all binary classification tasks with
> reported performance averaged over these multiple tasks.
>
> Ref1: Croce & Hein. Reliable evaluation of adversarial robustness with
> an ensemble of diverse parameter-free attacks. ICML 2020.
>
> Ref2: Zhang et al. Theoretically principled trade-off between robustness
> and accuracy. International conference on machine learning. ICML 2019.
>
> **Q3:** Adding ablation studies with varying batch sizes to demonstrate
> the approximation in Section 4.2.
>
> **A:** We appreciate the reviewer's constructive comment. We have
> conducted some empirical studies to investigate the proposed AdAP_LN's
> sensitivity to batch sizes on CIFAR10 dataset. The results are included
> in Figure 1 in the global response (PDF file). The results show that our method does
> not require a very large batch size to achieve a good performance and is
> generally not sensitive to batch size.
>
> **Q4:** it seems the authors forgot to define AdAP_LPN properly. The
> only clue I can find is at line 337, where AdAP_LPN is combining
> listwise and pointwise adversarial regularization, then in contrast
> AdAP_LN is probably just listwise regularization.
>
> **A:** We apologize for the confusion. The AdAP_LPN is defined as
> Equation 10 in Section 4.1. Please refer to line 194.
>
> **Q5:** About the threat model behind TRADES, and PGD, confirm whether
> these threat models (models used to generated adversarial attacks) are
> the same as the AdAP family models or not.
>
> **A:** As introduced in Sec 5.1 and 5.2, when we evaluate the robustness
> against white-box attacks, the threat model is the evaluated model
> itself since white-box attacks have full access to the target model.
> However, when we evaluate the robustness against black-box attacks, the
> threat model behind TRADES, PGD, MART and AdAP family models is the same
> model trained with CE loss minimization on clean data for fair
> comparison.

---

### Official Review · Reviewer_axcz · 2023-07-10

**Soundness:** 3 good
**Presentation:** 3 good
**Contribution:** 3 good
**Rating:** 7
**Confidence:** 2

**Summary:**

This paper investigates how to improve the robustness of a model under adversarial attacks while ensuring its Average Precision (AP) on clean data samples. This studied problem can be very important in some application scenarios but has not been extensively explored yet. By integrating the idea of existing adversarial training based methods into AP maximization algorithms, a novel solution is proposed in this paper. Experimental results obtained on multiple datasets with various binary imbalanced settings demonstrate the superiority of the proposed solution in terms of AP and robustness, comparing with baseline methods, which only focusing on optimizing either AP or robustness of models.


**Strengths:**

1. The problem explored in this paper, i.e., enhancing model robustness while maintaining AP, is a practical and important problem in some application scenarios but has not been well studied, as related works only focused on improving either model robustness or model AP.
2. A novel solution is proposed in this paper by integrating existing adversarial training methods with AP maximization algorithms. Experimental results including performance comparison and ablation studies verify the effectiveness of the proposed solution.

**Weaknesses:**

1. To provide a more comprehensive perspective to evaluate the robustness of trained models, stronger attack methods, such as AutoAttack, should be included in experiments.
2. Authors didn't discuss limitations of the proposed solution. Based on the description in the methodology part, the training efficiency of the proposed solution may be a problem.

**Questions:**

1. In the introduction part, authors mentioned one nice property when designing adversarial training methods for AP maximization is "consistency of attacks between training and inference". Can authors add more explanation about this claim? Why attacks are expected to be consistent in training and inference? Generally, to evaluate model robustness more comprehensively, any kinds of attacks can be used in the inference stage.
2. How's the training efficiency of the proposed solution? It seems the proposed solution will take much longer time on training, comparing with adversarial training based methods.

**Limitations:**

The efficiency of the proposed solution may be a problem. Hence, it would be better if authors can discuss or provide some experimental results to explain the efficiency of the proposed solution.

---

> ### Author Rebuttal · Authors · 2023-08-10
>
> We thank the reviewer for the constructive comments.
>
> **Q1:** To provide a more comprehensive perspective to evaluate the
> robustness of trained models, stronger attack methods, such as
> AutoAttack, should be included in experiments.
>
> **A:** We appreciate the reviewer's suggestion. To provide a more
> comprehensive assessment, we have evaluated all the models again a
> strong attack method $Auto-PGD_{CE}$ proposed in \[Ref1\]. The results
> are summarized in Table 1 in the global response(PDF file). We can observe that $Auto-PGD_{CE}$ exhibits a
> stronger attack as it leads to the deterioration of performance across
> all models, compared with the iterative FGSM (standard PGD) method
> employed in section 5.1. However, we can see that the superiority of our
> proposed methods still remains evident. Please note that we did not
> compare with the ensemble AutoAttack since it is tailored for
> multi-class classification concerning accuracy. For instance, the
> $APGD_{DLR}$ approach presented in \[Ref1\] requires at least three
> classes. We focus on AP maximization on imbalanced binary classification
> data and the multi-class datasets such as CIFAR-10 are converted into
> multiple one-vs-all binary classification tasks with reported
> performance averaged over these multiple tasks.
>
> Ref1: Croce & Hein. Reliable evaluation of adversarial robustness with
> an ensemble of diverse parameter-free attacks. ICML 2020.
>
> **Q2:** Authors didn't discuss limitations of the proposed solution.
> Based on the description in the methodology part, the training
> efficiency of the proposed solution may be a problem.
>
> **A:** We thank the reviewer for the constructive comment. We agree with
> the reviewer that the proposed adversarial training methods are more
> time-consuming than conventional natural training and some adversarial
> training methods (e.g. PGD). For detailed runtime analysis, please refer to
> the response to Q4. We will add a limitation part to discuss this in the
> revision.
>
> **Q3:** Regarding the \"consistency of attacks between training and
> inference\", can authors add more explanation about this claim? Why
> attacks are expected to be consistent in training and inference?
>
> **A:** We apologize for the confusion. The consistency of attacks
> between training and inference means that the attack is generated only
> based on individual data. In the inference stage, the attack is usually
> imposed on individual data without referring to other data, which is
> referred to as pointwise attack. Hence, we propose to use the pointwise
> attack instead of listwise attack in the training stage to maintain such
> consistency, which is helpful for improving the results. It is important
> to note that we do not restrict the pointwise attacks used in the
> training to be the same as the inference. We have added an experiment to
> demonstrate the effectiveness of our approaches when the inference uses
> a different pointwise attack ($Auto-PGD_{CE}$ attack). The results are
> demonstrated in response to Q1.
>
> **Q4:** How's the training efficiency of the proposed solution? It seems
> the proposed solution will take much longer time on training, comparing
> with adversarial training based methods.
>
> **A:** Below, we've included the results of efficiency comparisons for all the models. In the experiment, we set the
> parameters, which could affect training time, exactly the same(e.g.
> batch size as 128, total epochs as 60, adversarial samples are generated
> with 6 projected gradient ascent steps) and run all the models over
> three times on the Class_0 task of CIFAR10. From the table, we can
> observe that (i) adversarial training methods are generally more
> time-consuming than natural training; (ii) our proposed AdAP_LN and
> AdAP_LPN methods cost a little more time than traditional PGD method but
> much less time than TRADES. This is because, to generate adversarial
> samples in training, TRADES is solving the maximization of KL divergence
> between the probabilities predicted with clean data and perturbed
> data(i.e.
> $\max_{\|\delta\|\leq \epsilon}\sum_k h(x)_k\log h(x)_k - h(x)_k\log h(x+\delta)_k$,
> where $h(x)_k$ and $h(x+\delta)_k$ are predicted probabilities for class
> k on clean data and perturbed data respectively ) while PGD and our proposed methods are directly solving maximization of Cross Entropy
> (i.e. $\max\_{ \|\delta\| \leq \epsilon }-\log h(x+\delta)_y$). In adversarial training, since each gradient descent step wrt $w$ requires multiple gradient ascent steps wrt $\delta$, the
> computational expense primarily stems from the
> projected gradient ascent steps, which can be also observed by comparing the efficiency of CE Min. with PGD.
> And TRADES requires two
> forward propagations and one backpropagation in each projected gradient
> ascent step, whereas the latter method only needs one forward
> propagation and one backpropagation.
>
> | Methods   | Run 1 | Run 2 | Run 3 | Average |
> |-----------|---------|---------|---------|-----------|
> | CE Min.   | 563s    | 566s    | 568s    | 565.67s   |
> | AP Max.   | 589s    | 590s    | 589s    | 589.33s   |
> | PGD       | 2833s   | 2804s   | 2803s   | 2813.33s  |
> | TRADES    | 4203s   | 4182s   | 4179s   | 4188.00s  |
> | MART      | 3192s   | 3205s   | 3194s   | 3197.00s  |
> | AdAP\_LN  | 3227s   | 3213s   | 3211s   | 3217.00s  |
> | AdAP\_LPN | 3234s   | 3219s   | 3218    | 3223.67s  |

---

> > ### Comment · Reviewer_axcz · 2023-08-20
> > **Thank you for your reply**
> >
> > I have read authors' reply, which addressed most of my questions and concerns. Considering the overall quality of this work, I decide to keep my original score 7.

---

### Author Rebuttal · Authors · 2023-08-10

We thank the reviewers for your comments and feedback on our paper. We have included some experimental results in the PDF file, including adversarial robustness against $Auto-PGD_{CE}$ white-box attack, adversarial accuracy against white-box iterative FGSM attack on CIFAR10 dataset, and illustration for insensitivity to batch size of AdAP\_LN.

---

### Decision · Program_Chairs · 2023-09-21

**Decision:**

Accept (spotlight)

**Comment:**

AC browse through the paper and finds the problem interesting, which is to optimize for AP while training for adversarial robustness. This problem statement seems novel. Ratings are consistent across the board to accept.